# COOL: CHAIN-ORIENTED OBJECTIVE LOGIC WITH NEURAL NETWORKS FEEDBACK CONTROL FOR DYNAMIC MULTI-DSL REGULATION

## ABSTRACT

Multi-DSL regulation requires dynamic coordination of modular domain logic, yet existing frameworks lack mechanisms for cross-DSL management. We address these challenges with **COOL (Chain-Oriented Objective Logic)**, a neural-symbolic framework that introduces: (1) **Chain-of-Logic (CoL)**: Structures the reasoning process into hierarchical, expert-guided multiple sub-DSLs with heuristic vectors and runtime keywords; and (2) **Neural Network Feedback Control (NNFC)**: A self-correcting mechanism that isolates neural components into reusable libraries, filtering erroneous predictions via sequential network coupling. Through rigorous theoretical analysis, we formally establish the efficacy of COOL under dynamic situation. Ablation studies on relational and symbolic tasks validate that: CoL achieves 70% higher accuracy than the baseline variant without CoL while reducing computational overhead by **91% fewer tree operations** and **95% faster reasoning**. Under adversarial conditions—insufficient training data, increased complexity, and multi-library requirements—NNFC further improves accuracy by **6%** and reduces tree operations by **64%** compared to the CoL-only variant. Both theoretical analysis and experimental validation confirm COOL as a highly efficient and reliable framework for multi-DSL regulation. (need reversion)

## 1 INTRODUCTION

Domain-Specific Languages (DSLs) are pivotal for encapsulating domain logic in fields ranging from robotics to finance, enabling efficient and verifiable system design. The regulation of multiple modular DSLs offers a compelling path toward scalable and reusable system composition, surpassing the limitations of monolithic designs. However, multi-DSL regulation introduces a fundamental paradox: effective collaboration necessitates the interleaved application of rules from heterogeneous modules, yet this very interaction inevitably provokes the fundamental risk of unregulated rule application behaviors that undermine system safety and reasoning efficiency. This core tension between a necessary mechanism and its detrimental consequences remains unresolved, placing a substantial burden on programmers who must manually write and maintain intricate regulation code.

The root cause of this paradox lies in the misalignment between rule premises and their applicable domains during multi-DSL collaboration. A rule's premise and specific pattern matching, sufficient within its native DSL, no longer guarantee correct application in a broader, heterogeneous context. Manual coordination via intermediate code attempts to resolve this by realigning each DSL to the new problem domain—a process that requires extensive debugging, even necessitates modifications to the original DSLs themselves, and remains prone to errors. This process ultimately aims to reconfigure the application scope of rules to prevent unregulated rule application behaviors. Consequently, the problem can be reframed: **directly and dynamically governing rule application scope offers a principled alternative to manual realignment**.

Our approach to multi-DSL regulation, COOL (Chain-Oriented Objective Logic), intersects with several research areas, yet remains fundamentally distinct from them: Work in *DSL modulations* focuses on structuring domain logic but lacks mechanisms for dynamic runtime cross-DSL coordination. Research in *neural adaptive control* (e.g., in robotics or process control) offers learning-based dynamic adaptation but is confined to the continuous state spaces of physical systems, making it

inapplicable to the discrete, symbolic reasoning required for coordinating multiple DSLs. Techniques in *neural-guided search* effectively compress search spaces but are fundamentally designed for static, monolithic situations and cannot handle the dynamic, heterogeneous nature of multi-DSL regulation.

To address these limitations, COOL—the first framework specifically designed for dynamic multi-DSL regulation, introduces two novel components that work in concert to precisely and dynamically govern rule application scope and resolve conflicts:

(1) Chain-of-Logic (CoL) dynamically defines the affiliation of rules to specific DSLs through heuristic vectors. Its runtime keywords act as transition operators, dynamically switching between or terminating DSLs based on contextual states, thereby preventing conflicting applications across modules.

(2) Neural Network Feedback Control (NNFC) complements CoL by introducing lightweight, modular neural agents. The number of these agents automatically scales with the number of DSLs. They continuously monitor the regulation process and adaptively refine the scope of the rules they are responsible for, mitigating the instability risks inherent in neural components by filtering erroneous predictions.

By integrating CoL's symbolic precision with NNFC's adaptive robustness, COOL effectively resolves the paradox inherent in multi-DSL regulation.

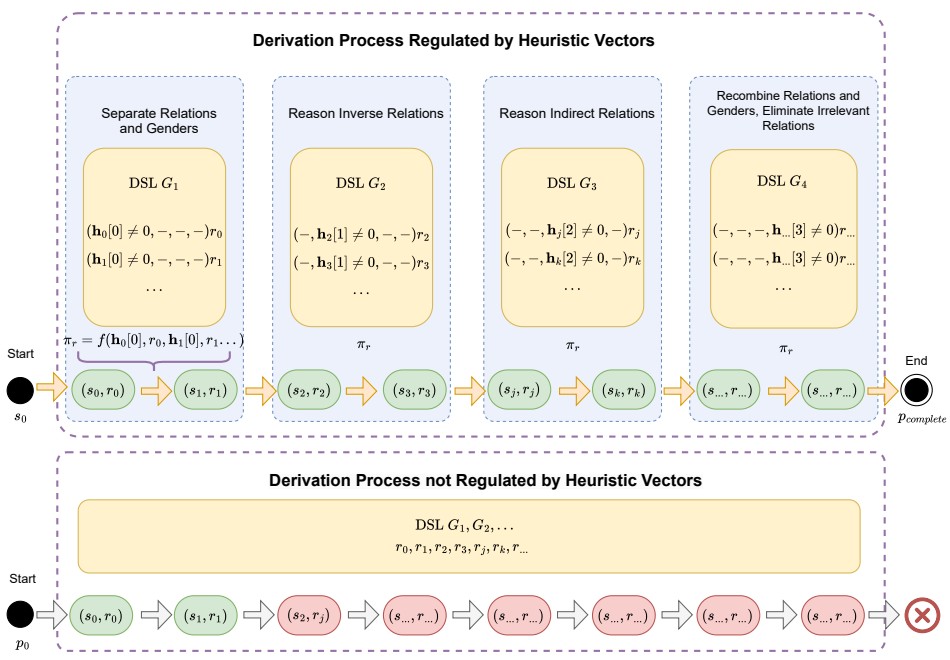

Figure 1: Regulation of DSLs **G** in Chain-of-Logic.

To rigorously validate the COOL framework, we integrate theoretical analysis with empirical ablation studies. Theoretically, we establish its foundation through formal methods: for CoL, we prove its expressiveness in encapsulating multi-DSL logic and derive complexity bounds for its regulation mechanism, demonstrating its efficacy in suppressing unregulated behavior emergence and confining the search space; for NNFC, we formalize its feedback control and filtering mechanism, proving its convergence properties and analyzing stability under adaptive conditions using Lyapunov analysis. Experimentally, we conduct extensive *internal ablation studies* within the COOL framework under both static (fixed conditions) and dynamic (evolving conditions) scenarios. Results show that the CoL component alone significantly improves accuracy by 70%, while reducing tree operations by 91% and time by 95% compared to a baseline without CoL's declarative control primitives. In dynamic experiments, NNFC further boosts accuracy by 6% and cuts tree operations by 64% compared to the CoL-only variant. Together, these theoretical and empirical results conclusively validate COOL as

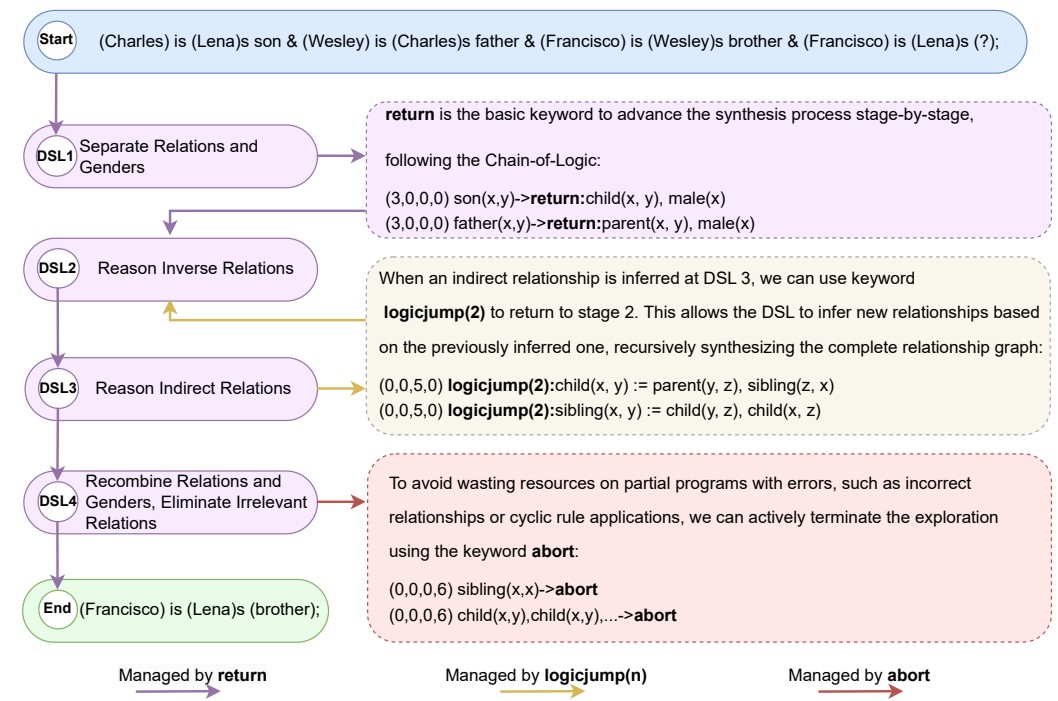

Figure 2: Keywords in Chain-of-Logic.

the first framework enabling dynamic multi-DSL regulation through the synergistic combination of CoL's declarative control flow and NNFC's adaptive modulation.

The contributions of our work are as follows:

1. We propose the **Chain-of-Logic (CoL)**, a structured paradigm for multi-DSL regulation that uses **heuristic vectors** to dynamically define rule affiliations, with runtime **keywords** acting as transition operators to govern cross-DSL control flow—collectively suppressing unregulated behavior emergence

2. We introduce **Neural Network Feedback Control (NNFC)**, a self-correcting mechanism that employs lightweight modular neural agents to adaptively refine scoping boundaries and filter erroneous predictions, ensuring stability.

3. We establish a theoretical and empirical foundation for multi-DSL regulation, proving the properties of CoL and NNFC and validating their integration in **COOL**.

## 2 METHOD

This section details the implementation of the COOL framework, focusing on the principles of its two core components designed for multi-DSL regulation: the Chain-of-Logic (CoL) and Neural Network Feedback Control (NNFC). Our approach operates on Domain-Specific Languages (DSLs), formally defined as grammatical formalisms (including but not limited to context-free grammars), where the core reasoning process involves the step-wise derivation of symbolic structures from an initial nonterminal through the sequential application of production rules (Appendix C.2).

### 2.1 CHAIN-OF-LOGIC (COL)

The Chain-of-Logic (CoL) provides a control flow paradigm to regulate multiple homogeneous DSLs that require frequent switching and rule sharing across multi-step tasks. This is achieved through two

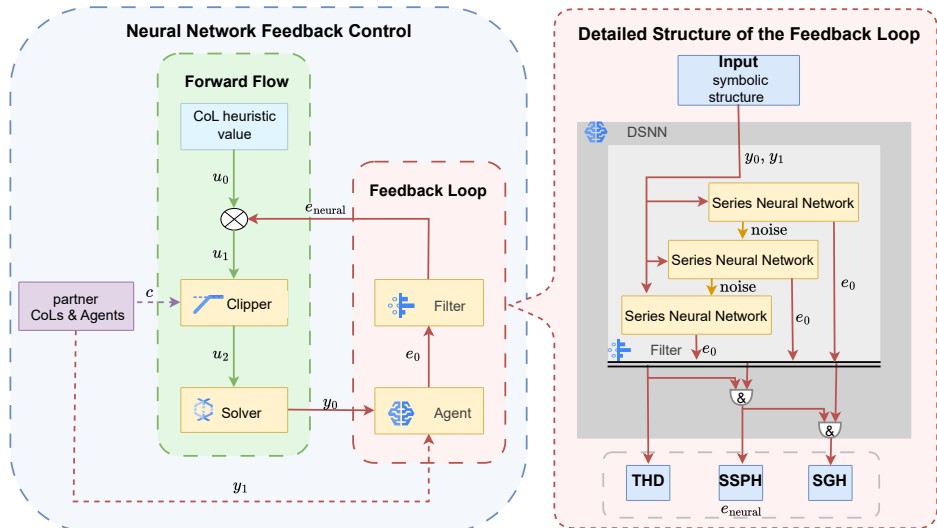

Figure 3: Neural Network Feedback Control. The left side illustrates the complete control loop of NNFC. In the forward flow (green path), heuristic values $u$ guide the synthesis process as control signals. In the feedback loop (red path), the neural agent generates initial error signals $e_0$ from partial programs $y$. These singals are then filtered to produce high-quality error signals $e_{\text{neural}}$, which adjust the initial heuristic values $u_0$. In multidomain synthesis, the CoL and agent from the self-domain use partner domain information (dashed path) to clarify tasks and avoid competition, ensuring modularity. The right side details the feedback loop: The agent comprises multiple neural networks coupled in series via noise signals, with each network generating its own error signal $e_0$, then these signals with large discrepancies are filtered, retaining the final high-quality error signals $e_{\text{neural}}$. (See Appendix D for details on the neural network's format, architecture, training, prediction, and role in synthesis.)

core mechanisms: heuristic vectors, which define rule affiliations, and their runtime manifestations, keywords, which govern cross-DSL control flow transitions.

**heuristic vectors.** As shown in Figure 1, a heuristic vector dictates the workflow sequence for applying multiple DSLs to a multi-step task. Each rule is not exclusively owned by a single DSL but is instead annotated by a heuristic vector. The positions within the vector correspond to the ordered DSLs in the workflow; only DSLs at positions with non-zero values possess the permission to utilize that rule. For example, in Figure 5, a rule with the heuristic vector $(0, 7, 3)$ can be used by the second and third DSLs in the sequence. The specific non-zero values (7 and 3) provide heuristic guidance for the rule's application within their respective DSLs' control mechanisms. During the collaborative derivation process regulated by CoL, a symbolic structure is processed streamingly along the predefined DSL workflow. The process can skip downstream DSLs but cannot revert to upstream ones.

**keywords.** CoL introduces three keywords—`return`, `logicjump(n)`, and `abort`—which dynamically manage workflow progression within and across DSLs during regulation (Figure 2):

1. **return**: Ends the current rule application, staying within the current DSL or advancing to subsequent DSLs.

2. **logicjump(n)**: Jumps directly to DSL n, enabling non-sequential transitions across the DSL workflow.

3. **abort**: Terminates the current derivation branch.

## 2.2 NEURAL NETWORK FEEDBACK CONTROL (NNFC)

Neural Network Feedback Control (NNFC) complements the CoL framework by integrating an adaptive neural component, enabling continuous learning and refinement of the multi-DSL regulation process. This capability not only enhances collaboration among homogeneous DSLs when constrained

to a single CoL instance, but also enables coordination of heterogeneous DSLs when extended across multiple CoL instances in cross-task scenarios.

NNFC operates by maintaining a dedicated neural agent for its associated CoL. This agent learns from historical regulation trajectories of the CoL to dynamically refine future rule applications by adjusting their heuristic guidance values, thereby scoping their applicability. As illustrated in Figure 3, NNFC's architecture comprises:

**(1) Forward Flow:** CoL signals, based on rule application policies, are combined with the feedback signals from the neural agent. The Clipper module then prioritizes control signals consistent with the neural agent's guidance by capping inconsistent signals (Appendix E). Finally, the solver applies rules based on the refined signals and generates new symbolic structures.

**(2) Feedback Loop:** The neural agent generates refinement signals from intermediate symbolic constructs. To suppress the impact of mispredictions, the Filter module refines these signals before they influence the forward flow.

The neural agent is central to NNFC. Its structure (Figure 3, right) employs multiple sequentially connected networks with identical architectures—a design choice that simplifies analysis and ensures the induced representation divergence stems solely from the serial data flow rather than architectural differences. However, as each subsequent network processes both the original input and the transformed outputs from its predecessor, their effective input structures and processing paths diverge. This serial design not only enables consistency checks by actively inducing divergent internal representations of the same initial input, but also amplifies inconsistencies in discordant outputs (Appendix D.4). The agent leverages the filtered signals to fine-tune the forward flow (Appendix **??**).

The neural agent generates three types of outputs that refine the multi-DSL regulation process. Figure 4 illustrates how these outputs enhance the regulatory workflow (see Table 5 for detailed features):

1. **Task Detection Head (TDH)**: Determines whether the current symbolic structure falls within the processing scope of the CoL framework in cross-task scenarios.

2. **Search Space Prune Head (SSPH)**: (Active when TDH is true) Assesses the feasibility of deriving a target symbolic structure from the current state, enabling search space pruning by eliminating infeasible paths.

3. **Search Guidance Head (SGH)**: (Active when both TDH and SSPH are true) Outputs features of promising rules—including the affiliated DSL, premise symbolic structure, and heuristic value—to guide the application of specific rules.

## 3 THEORETICAL ANALYSIS

We establish the theoretical foundation of COOL through formal analysis of its core components: (1) Chain-of-Logic (CoL)'s expressiveness in multi-DSL encapsulation and complexity bounds for its regulation mechanism; (2) Neural Network Feedback Control (NNFC)'s convergence properties and stability guarantees under adaptive conditions. Our proofs demonstrate that COOL provides a principled solution for governing multi-DSL regulation, ensuring both efficacy and reliability.

### 3.1 CHAIN-OF-LOGIC (COL)

**CoL preserves the full expressive power of regulated DSLs.** The composite framework's expressiveness equals that of its most expressive constituent DSL, as CoL introduces only finite-state coordination mechanisms without adding unbounded computation. Thus, the system inherits the capabilities of any constituent DSL (finite-state, context-free, or Turing-complete), ensuring CoL acts as a transparent coordinator rather than a constraint. (Proof A.1)

**CoL's regulatory expressiveness—defined by the class of regulatory strategies it can enact—adaptively scales with the governed DSLs' computational hierarchy.** The control flow paradigm enables sophisticated regulation strategies that suppress non-regulated applications and enhance efficiency, dynamically parameterizing keywords using DSL outputs. Specifically, CoL's regulatory expressiveness scales correspondingly through finite-state, context-free, and Turing-complete

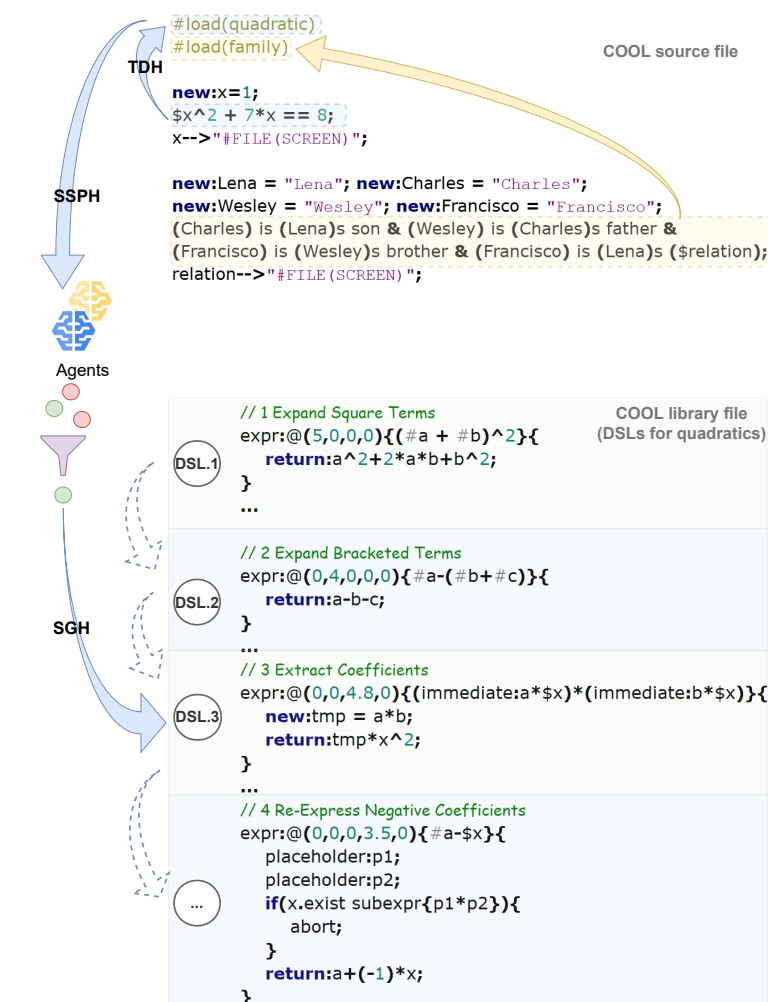

Figure 4: How Neural Networks' Multi-Head Outputs Function. When a specific DSL library is loaded, its corresponding neural agent (a specialized module) is loaded to assist in multi-DSL regulation by influencing rule application policies for intermediate states and their derivatives. For instance, if the neural agent bound to the "quadratic" DSL reads an intermediate state containing a quadratic equation (blue background), it: (1) guides the coordinator to apply rules specific to its DSL via TDH, (2) prioritizes applying rules on intermediate states that are easier to derive using SSPH, and (3) directs the coordinator to apply specific rules through SGH to progress toward a specified CoL activity. Simultaneously, it prevents the coordinator from applying quadratic DSL rules to unrelated tasks (e.g., yellow background), ensuring no interference with other DSL reasoning processes. These functions, achieved by modifying heuristic values of DSL-specific rules, demonstrate strong modularity.

levels based on its governed DSLs' capabilities, ensuring effective coordination across computational hierarchies without imposing predefined constraints. (Proof A.2)

**Complexity Analysis.** CoL systematically reduces complexity via three algorithm-agnostic mechanisms governed by keywords. Baseline complexity without regulation is $O\left(R^{L_{\text{global}}}\right)$ for global rule set size $R$ and unguided path length $L_{\text{global}}$. CoL reduces this through: (1) `return` and `logicjump(n)` dynamically restricting the active rule set, reducing effective branching factor $b_t$ at step $t$; (2) `abort` pruning branches with multiplicative factor $(1 - p_t)$ at step $t$.

The composite complexity is $O\left(\prod_{t=1}^{L}\left[(1 - p_t) \cdot b_t\right]\right)$ for regulated path length $L$ (Section C.3), bounded between worst-case $O\left(R^L\right)$ (disabled regulation, Proof C.4.1) and best-case $\Omega(L)$ (perfect

regulation, Proof C.4.2). Practically, CoL's exponential reduction (via branching factor restriction) and multiplicative reduction (via pruning) compound multiplicatively to suppress unregulated search expansion (Section C.5), with efficacy depending on heuristic vector and keyword precision. (Proof A.3)

## 3.2 NEURAL NETWORK FEEDBACK CONTROL (NNFC)

**Heuristic values and neural oracle information can be safely composited without compromising algorithmic completeness, while performance improvement depends on oracle accuracy.** For algorithms satisfying completeness and optimality, a global method modifies heuristic values based on oracle information to retain these properties if the oracle is correct. Under resource constraints, compositing generally increases the probability of finding a solution or reduces expected time, but erroneous predictions diminish improvement. (Proof B.1)

**The series coupling of neural networks amplifies inconsistencies via error accumulation, enhancing filtering sensitivity with probability proportional to the neural error rate $\epsilon$.** This series structure produces larger inconsistency indicators than parallel setups due to error propagation, particularly in low-error regimes. The filtering probability follows $p_{\text{filter}} \propto \beta \cdot \epsilon \cdot \frac{1-\gamma^K}{1-\gamma}$, where $\gamma$ is the error transfer coefficient and $K$ is the number of series networks, ensuring adaptive filtering that strengthens with increasing error rates. (Proof B.2)

**System stability is governed by dynamic error control and filtering, converging to a bounded region with steady-state error proportional to dynamic error over filter efficacy.** Using the energy function $V = V_{\text{neural}} + V_{\text{output}}$ where $V_{\text{neural}} = \|\pi^* - \pi_{\text{neural}}\|^2$ and $V_{\text{output}} = \|\pi^* - \pi_{\text{output}}\|^2$, the energy change $\Delta V$ depends on training effectiveness, dynamic error $e_{\text{dynamic}} = e_{\text{drift}} + e_{\text{forget}}$, and filtering. Stability requires $\Delta V \leq 0$, and from the analysis of $V_{\text{output}}$, the steady-state error is given by $e_{\text{steady}} \approx \frac{\beta e_{\text{dynamic}}}{\alpha \kappa}$ for small dynamic error, where $\alpha$ is the learning efficiency coefficient, $\beta$ is the dynamic error injection coefficient, and $\kappa$ is the sensitivity coefficient of the termination rate. Filtering ensures a minimum termination rate $\rho_{\text{min}} > 0$, which maintains learning opportunities and prevents divergence when neural error is high. (Proof B.3)

## 4 EXPERIMENTS

We evaluate COOL through complementary phases: *static experiments* under fixed conditions isolate CoL's contribution to multi-DSL regulation, while *dynamic experiments* with varying conditions assess NNFC's adaptive capabilities. This structured approach directly reflects our theoretical framework.

Table 1: Benchmark configurations. Relational benchmarks are divided into easy and difficult groups based on the number of relationship edges, while symbolic benchmarks are based on the number of nodes in the tree.

| Benchmark Type | Difficulty Level A | Difficulty Level B |
|---|---|---|
| relational | 300 tasks with 3 edges | 200 tasks with 4 edges |
| symbolic | 300 tasks with around 5 nodes | 200 tasks with around 9 nodes |

### 4.1 EXPERIMENTAL SETUP

**Environment.** The COOL framework's unique requirements—including heuristic vector management, keyword-driven control flow, and neural feedback loops—demand dedicated language support for proper implementation. We therefore designed and implemented the COOL host language to natively support these capabilities. All experiments were conducted on a system equipped with an Intel i7-14700 CPU, GTX 4070 GPU, and 48GB RAM. (Language specifications and design rationale are detailed in Appendix C.)

**Benchmarks.** Our experiments include both relational and symbolic tasks of varying difficulty, as summarized in Table 1 (see Appendix F for detailed examples). Specifically, the *relational*

tasks require reasoning about target relationships through common-sense rule applications. The *symbolic* tasks entail solving non-standard quadratic equations using manual calculation steps with mathematical laws.

**Metrics.** We evaluate COOL's multi-DSL regulation efficiency using: **(1) Accuracy**: proportion of correctly regulated reasoning tasks under resource constraints.[1] **(2) Tree Operations**: number of rule applications required for regulated tasks. **(3) Memory Overhead**: number of transformation pairs (an intermediate symbolic structure paired with the rule to be applied. Appendix C.2) **(4) GPU Overhead**: number of neural network calls. **(5) Time Overhead**: actual time spent on multi-DSL regulation tasks.

Table 2: Static performance of DSL and CoL DSL for relational and symbolic tasks. CoL DSL significantly outperforms DSL in all metrics.

| Benchmark | Group | Accuracy↑ (%) | Avg. Tree Operation↓ | Avg. Transfor- mation Pair↓ | Avg. Time Spent↓(s) |
|---|---|---|---|---|---|
| relational | DSL | 11.3 | 463.9 | 1432.2 | 9.43 |
| | CoL DSL | **100.0** | **46.6** | **177.8** | **0.48** |
| symbolic | DSL | 48.3 | 411.2 | 2285.3 | 3.31 |
| | CoL DSL | **100.0** | **33.8** | **92.7** | **0.11** |

**Groups.** Our ablation study evaluates two primary dimensions: the CoL regulatory mechanism (tested in static experiments by comparing regulation performance with and without CoL) and the NNFC mechanism (tested in dynamic experiments with and without neural network feedback control), while heuristic vectors and filter serve as secondary variables to isolate individual component contributions. (See Appendix F.2 for detailed group configurations.)

## 4.2 STATIC EXPERIMENTS

We first evaluate CoL under fixed conditions (domain, difficulty, and pre-trained neural networks) to isolate its impact. Controlled experiments confirm CoL's systematic improvements across all metrics.

The results in Table 2 demonstrate that **CoL significantly enhances multi-DSL coordination accuracy while minimizing overhead**. CoL improves accuracy from below 50% to 100% across both relational and symbolic benchmarks. It also achieves substantial reductions: in relational tasks, tree operations drop by 90%, state transitions by 88%, and time by 95%; in symbolic tasks, tree operations decrease by 92%, state transitions by 96%, and time by 97%. These results highlight CoL's efficacy in optimizing multi-DSL regulation metrics, **confirming the complexity reduction theorized for CoL in our theoretical analysis**.

Secondary ablation and extension experiments clarify and prove:

**First, global heuristics enhance multi-DSL regulation efficiency, and Heuristic Vector enables hierarchical rule guidance for better regulation.** As shown in Figure 7, CoL (Heuristic) outperforms baseline Direct in most metrics, and CoL significantly surpasses CoL (Heuristic) in all metrics, indicating CoL's positive impact by structuring rule application. Explicit activity decomposition yields greater performance gains.

**Second, integrating CoL with neural networks further improves search efficiency, validating theoretical composability.** CoL + NN reduces tree operations by 43% and state transitions by 19% in relational tasks, and by 64% and 46% in symbolic tasks, showing neural networks further narrow the search space. Inner coupling is essential for reliability, consistently outperforming non-neural groups and enhancing robustness through error filtering.

**Third, filtering structure is more effective when error tolerance is low, demonstrating CoL's greater utility in non-completeness-oriented coordination tasks.** For symbolic tasks, groups with inner coupling outperform those without, while for relational tasks with higher error tolerance,

---

[1]Resource limit: Maximum 1000 rule applications and path length 50.

filtering may incur a net penalty due to loss of valid outcomes. Thus, CoL excels in tasks where errors are critical.

## 4.3 DYNAMIC EXPERIMENTS

Table 3: Dynamic performance of CoL DSL and CoL DSL+NNFC(Flt). NNFC significantly improves the dynamic performance of CoL DSL in accuracy, tree operations, and transformation pairs.

| Bench-mark | Group | Accuracy↑ (%) | Avg. Tree Operation↓ | Avg. Trans-formation Pair↓ | Avg. Neural Network Invocation↓ | Avg. Time Spent↓(s) |
|---|---|---|---|---|---|---|
| relational | CoL DSL | **100.0** | 70.0 | 259.8 | **0** | **1.05** |
| | CoL DSL+NNFC (Flt) | **100.0** | **54.6** | **224.5** | 21.7 | 2.08 |
| symbolic | CoL DSL | 82.6 | 233.5 | 977.1 | **0** | 1.42 |
| | CoL DSL+NNFC (Flt) | **99.4** | **50.3** | **222.2** | 21.6 | **1.12** |
| multi-domain | CoL DSL | 97.5 | 115.2 | 367.6 | **0** | **0.99** |
| | CoL DSL+NNFC (Flt) | **99.0** | **45.6** | **250.5** | 72.84 | 3.91 |

We evaluate NNFC under dynamic conditions (varying domains, difficulty, and evolving neural networks) to assess its adaptability. Results confirm that NNFC significantly enhances reliability, sustaining high accuracy in real-world scenarios while reducing computational overhead despite increasing task complexity.

The results in Table 3 confirm that **NNFC compensates for CoL's deficiency in dynamic alignment during dynamic regulation scenarios, thereby validating stability guarantees.** As task difficulty increases and cross-task heterogeneous DSL regulation scenarios emerge, the accuracy of the CoL group declines compared to static experiments. However, the NNFC-enhanced group maintains at least 99% accuracy, demonstrating strong adaptivity. Additionally, it reduces tree operations by 22% and state transitions by 14% compared to CoL alone. For symbolic tasks, despite added neural network invocation time, NNFC reduces overall time by 21%.

Further ablation experiments confirm that:

**Filtering is indispensable for NNFC's stability.** (Appendix F.4) As shown in Figures 8 and 9, filtering reduces misprediction-induced accuracy declines by 94%.

When a neural agent underperforms due to insufficient training data (Figure 8, tasks 51-100), inadequate generalization to challenging tasks (Figure 8, tasks 301-350), or catastrophic forgetting in new domains (Figure 9, tasks 1-100), incorrect predictions cause the regulation path to deviate from CoL, reducing efficiency and accuracy. With filtering, the attenuation ratio spikes, indicating extensive prediction filtering, which ensures adherence to CoL and mitigates negative impacts. As the neural agent stabilizes (Figure 8, tasks 101–300, 351–500; Figure 9, tasks 101–400), the attenuation ratio decreases, reflecting reduced filtering need and increased agent influence on regulation policy. This adaptive process maintains efficiency and robustness. **This demonstrates the necessity of filtering for ensuring stability, validating our theoretical analysis.**

## 5 CONCLUSION

We introduced COOL (Chain-Oriented Objective Logic) as the first framework specifically designed for dynamic multi-DSL regulation. We developed Chain-of-Logic (CoL) with its runtime keywords enabling expressive cross-DSL control flow and Neural Network Feedback Control (NNFC) providing adaptive realignment. This combination supports both homogeneous DSL coordination within single tasks and heterogeneous DSL collaboration across tasks. We rigorously validated COOL through theoretical analysis of its complexity reduction and stability properties, with extensive static and dynamic experiments confirming its efficacy. We believe our work provides a valuable foundation for broader research in multi-DSL regulation.

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

## A DETAILED THEORETICAL SECTION OF CHAIN-OF-LOGIC

### A.1 EXPRESSIVENESS PRESERVATION UNDER FINITE-STATE CONTROL

*Proof.* For a given Chain-of-Logic (CoL) framework with a finite set of DSLs, its introduction does not alter the fundamental computational expressiveness of the underlying Domain-Specific Languages (DSLs) it orchestrates. The composite system's expressiveness is precisely that of the most expressive constituent DSL.

Let a CoL framework be defined by a finite set of control states $Q_c = \{A_1, A_2, \ldots, A_m\}$. This finite control regulates a set of DSLs $\mathcal{D} = \{D_1, D_2, \ldots, D_k\}$. The state of the composite system is given by the tuple $(q_c, q_d, s)$, where:

- $q_c \in Q_c$ is the current CoL state,

- $q_d$ is the current state of the active DSL $D_{\text{active}} \in \mathcal{D}$,

- $s$ represents any auxiliary storage (e.g., stack, tape) used by $D_{\text{active}}$.

The proof proceeds by analyzing the state space and capabilities of this composite system relative to the capabilities of $\mathcal{D}$.

#### GENERAL PROOF FOR ALL CASES

The CoL controller $Q_c$ is finite by definition. The expressiveness of the composite system is determined by the product of the CoL state space and the state space of the underlying DSLs. Crucially, the CoL framework only influences the selection of the active DSL and the rule application within it through finite-state transitions; it does not provide its own unbounded storage mechanism.

Therefore, the overall expressiveness is bounded by:

$$\text{Expressiveness}(\text{CoL} + D) \leq \text{Expressiveness}(D)$$

Conversely, since the CoL framework can be configured to simply select and remain within a single DSL $D$ (by setting a single activity that directly invokes $D$'s semantics), it can trivially simulate $D$'s behavior:

$$\text{Expressiveness}(D) \leq \text{Expressiveness}(\text{CoL} + D)$$

By combining these two results, we conclude:

$$\text{Expressiveness}(\text{CoL} + D) = \text{Expressiveness}(D)$$

This equality holds regardless of whether $D$ is finite-state, context-free, or Turing-complete. The finite-state control of CoL acts as a transparent orchestrator, adding no computational power beyond what the regulated DSLs provide.

$\square$

### A.2 FORMAL PROOF OF CoL CONTROL EXPRESSIVENESS

**Definition A.1** (CoL Control Function). The CoL control function $\delta_c$ maps the current control state to the next state based on keyword directives and DSL state outputs:

$$\delta_c(c_i, K, \sigma) = \begin{cases} c_{i+l(\sigma)} & \text{if } K = \texttt{return} \\ c_{n(\sigma)} & \text{if } K = \texttt{logicjump}(n) \\ \emptyset & \text{if } K = \texttt{abort} \end{cases}$$

where $\sigma$ denotes the operational state output from governed DSLs, and $l(\sigma), n(\sigma)$ are parameterization functions. $n, i, l \in \mathbb{N}$.

The computational expressiveness of $\delta_c$ is equivalent to the maximum expressiveness of the parameterization functions $n(\sigma)$ and $l(\sigma)$.

*Proof.* We prove by cases over the computational hierarchy:

**Case 1: Finite-State Expressiveness**
If all governed DSLs generate finite-state outputs, then $\sigma \in \Sigma_F$ for finite $\Sigma_F$. Consequently, $n(\sigma), l(\sigma)$ map to finite ranges, making $\delta_c$ a finite-state transducer's control unit.

**Case 2: Context-Free Expressiveness**
If any DSL provides stack-aware outputs, $\sigma$ encodes stack configurations. Then $n(\sigma)$ can implement stack-dependent transitions, enabling $\delta_c$ to simulate a pushdown automaton's control unit.

**Case 3: Turing-Completeness**
If any DSL produces Turing-computable outputs, $\sigma$ represents tape states. Then $n(\sigma)$ becomes Turing-computable, allowing $\delta_c$ to simulate any computable control strategy.

In all cases, $\delta_c$'s expressiveness is strictly determined by the computational power of its parameterization functions, which derive entirely from the governed DSLs. $\square$

### A.3 FORMAL COMPLEXITY ANALYSIS OF THE CoL

#### FUNDAMENTAL CONCEPTS AND BASELINE COMPLEXITY

**Definition A.2** (Complexity Metric). The computational complexity of a synthesis process is characterized by the **branching factor** $b$ (number of candidate rules evaluated per step) and the **path length** $L$ (total number of rule applications). The worst-case time complexity is bounded by $O(b^L)$.

**Definition A.3** (Baseline Complexity). Without CoL regulation, the synthesizer operates on the entire rule set $R$. The baseline complexity is:

$$\text{Complexity}_{\text{baseline}} = O\left(|R|^{L_{\text{global}}}\right)$$

where $|R|$ is the size of the global rule set and $L_{\text{global}}$ is the path length without guidance.

#### CoL COMPLEXITY-REDUCTION MECHANISMS

The CoL framework reduces complexity through three orthogonal mechanisms that are independent of the underlying search algorithm.

**Lemma A.4** (Effect of `return`). *The* `return` *keyword triggers an increment of the starting index $i$, strictly reducing the active rule set from $\{R_i, ..., R_N\}$ to $\{R_{i+1}, ..., R_N\}$. This induces a monotonically non-increasing sequence of branching factors $b_t$.*

**Lemma A.5** (Effect of `logicjump(n)`). *The* `logicjump(n)` *keyword sets the starting index $i$ to $n$, redefining the active rule set.*

- *If $n > i$ (backward jump), $|R_{active}|$ decreases, reducing $b_t$.*

- *If $n < i$ (forward jump), $|R_{active}|$ increases to ensure completeness.*

**Lemma A.6** (Effect of `abort`). *The* `abort` *keyword prunes the current search branch. It introduces a pruning ratio $p_t$ ($0 \leq p_t < 1$) at step $t$, reducing the effective branching factor from $b_t$ to $(1-p_t) \cdot b_t$.*

#### COMPOSITE COMPLEXITY FORMULATION

The combined effect of all three mechanisms results in the total complexity:

$$\text{Complexity}_{\text{CoL}} = O\left(\prod_{t=1}^{L}[(1 - p_t) \cdot b_t]\right)$$

where:

- $b_t$ is determined by the history of `return` and `logicjump` operations

- $p_t$ is determined by the effectiveness of `abort` pruning

- $L$ is the path length in the regulated search space

ASYMPTOTIC BOUNDS

The worst-case complexity under CoL regulation satisfies:

$$\text{Complexity}_{\text{CoL}} \leq O\left(|R|^L\right)$$

*Proof.* In the worst case where CoL regulation is disabled ($i$ remains at 1) and pruning is ineffective ($p_t = 0$), we have $b_t = |R|$ for all $t$, recovering the baseline complexity. □

The best-case complexity under CoL regulation satisfies:

$$\text{Complexity}_{\text{CoL}} \geq \Omega(L)$$

*Proof.* In the ideal case, through precise application of `return`/`logicjump` that perfectly restricts $b_t$ and perfect pruning via `abort` that achieves $(1 - p_t) \cdot b_t = 1$ for all steps $t$, the product term reduces to $1^L = 1$. The synthesizer must still perform $L$ steps, yielding the lower bound $\Omega(L)$. □

The actual complexity achieved depends on the precision of the regulatory strategies, with the theoretical bounds providing guarantees for all possible cases.

# B  DETAILED THEORETICAL SECTION OF NEURAL NETWORK FEEDBACK CONTROL THEORETICAL ANALYSIS

The system input is a fixed heuristic value $h(s, a)$, and the circuit output is a strategy $\pi(a|s)$. Through a superposition algorithm, a new strategy $\pi_{\text{output}}$ is generated:

$$\pi_{\text{output}} = \pi_{\text{input}} + \lambda \cdot \pi_{\text{neural}}$$

where $\lambda$ is a mixing coefficient, $\pi_{\text{input}}$ is a base strategy derived from $h$, and $\pi_{\text{neural}}$ is the neural network output strategy. The algorithm satisfies: if a circuit strategy exists, modify the heuristic value to align with the circuit strategy; otherwise, do not apply the strategy ($\lambda = 0$).

The neural network is continuously trained using data from search trajectories. Each training step uses sufficient data to update parameters via gradient descent:

$$\theta_n = \theta_{n-1} - \eta \nabla L(\theta_{n-1})$$

where $L$ is the loss function.

## B.1  PROOF OF THE COMPOSABILITY OF HEURISTIC AND ORACLE EFFECTS

In heuristic algorithms, heuristic values (e.g., cost, weight, probability, or reward) guide the search process, while an oracle provides additional information (e.g., true cost or optimal action). This proof framework demonstrates that oracle information can be safely composed with heuristic values without compromising the algorithm's properties (completeness and optimality) and significantly improves performance under resource constraints. Additionally, we analyze the impact of errors in circuit predictions, thereby narrowing the system's dynamic analysis to the circuit's dynamics.

WHEN THE ALGORITHM SATISFIES COMPLETENESS AND OPTIMALITY

For any heuristic algorithm that satisfies completeness and optimality, there exists a global method to modify heuristic values based on oracle information such that the modified algorithm retains completeness and optimality, provided the oracle information is correct.

*Proof.* We classify heuristic values by their function and outline the modification method for each class:

1. **Cost Estimation** (e.g., in A* algorithm): The oracle provides information about the true cost. The heuristic value is modified with the goal of aligning it more closely with the true cost compared to the original heuristic value. This modification preserves admissibility, thus maintaining optimality. The search space remains unchanged, preserving completeness.

2. **Weight**: The oracle indicates a node is superior, and its weight is increased (e.g., via multiplicative scaling). Weight modification only adjusts priorities without excluding nodes, preserving completeness. Oracle guidance toward the optimal solution preserves optimality.

3. **Probability**: The oracle indicates a node's probability should increase, and the probability distribution is renormalized. Non-zero probabilities ensure all nodes are selectable, preserving completeness. Increased probability for optimal nodes preserves optimality.

4. **Reward**: The oracle indicates an action's reward should increase, and the reward value is adjusted. All actions remain available, preserving completeness. Oracle guidance toward optimal actions preserves optimality.

**Composability**: Oracle information may appear multiple times. Each modification is based on correct information, allowing multiple modifications to composite without conflict.

**Error Analysis**: If the circuit prediction is erroneous (i.e., oracle information is incorrect), optimality may be compromised, but completeness is generally preserved unless errors exclude all solutions.

- In cost estimation, overestimation may skip optimal paths; underestimation may explore invalid paths, but all nodes remain accessible, so completeness holds.

- In weight, probability, or reward modifications, erroneous information may misguide search, but as long as correct nodes are not entirely excluded, completeness holds.

Thus, under completeness and optimality, oracle composability preserves completeness but may impact optimality if errors occur. □

GENERAL CASE (RESOURCE CONSTRAINTS, NO GUARANTEE OF COMPLETENESS OR OPTIMALITY)

Under resource constraints (e.g., limited time or memory), the compositing of oracle information generally leads to better outcomes on average, i.e., increased probability of finding a solution or reduced expected time, but erroneous predictions may diminish this improvement.

*Proof.* Let $P_{\text{found}}$ be the probability of finding a solution without oracle, and $E[T]$ be the expected time steps without oracle. The oracle provides information with probability $\gamma$ and accuracy $1 - \epsilon$ (where $\epsilon$ is the error rate).

When the oracle is correct, the effective search space is reduced from $|S|$ to $|S'| = k|S|$ (with $0 < k < 1$), increasing the probability of finding a solution:

$$P'_{\text{found, correct}} = \frac{T}{k|S|}.$$

When the oracle is erroneous, the search may be misled to invalid regions, potentially decreasing the probability:

$$P'_{\text{found, error}} \leq P_{\text{found}}.$$

The overall probability of finding a solution is:

$$P'_{\text{found}} = (1 - \epsilon)P'_{\text{found, correct}} + \epsilon P'_{\text{found, error}}.$$

For small $\epsilon$, $P'_{\text{found}} > P_{\text{found}}$.

Similarly, for expected time: when the oracle is correct, time is reduced by a factor $\beta$ ($0 < \beta < 1$); when erroneous, time may increase by a factor $\beta_{\text{error}} > 1$. The overall expected time is:

$$E'[T] = E[T]\left[(1 - \gamma) + \gamma(1 - \epsilon)\beta + \gamma\epsilon\beta_{\text{error}}\right].$$

For small $\epsilon$, $E'[T] < E[T]$.

**Error Analysis**: Erroneous circuit predictions weaken the search process, reducing the probability of finding a solution or increasing expected time.

**Mutual Conclusion**: The quality of circuit predictions directly determines search effectiveness: accurate predictions improve search, while erroneous predictions weaken it. The final search outcome and circuit predictions are interdependent: successful search provides positive feedback, reinforcing accurate predictions; failed search provides negative feedback, correcting erroneous predictions. Therefore, dynamic analysis of the entire system can be confined to the circuit's dynamics, as the circuit's performance (accuracy and error rate) is the key factor governing search behavior. $\square$

CONCEPTUAL PROOF: HOMOGENEITY AND HARMONIOUS COMPOSABILITY

**Lemma B.1** (Homogeneity). *Heuristic values and oracle information are homogeneous, as both derive from historical data or predictive models. Heuristic values are derived from past experience, while the oracle provides immediate feedback.*

**Lemma B.2** (Harmonious Composability). *Due to homogeneity, heuristic values and oracle information are compatible. There always exists a compositing method that comprehensively integrates both types of information. The composite effect is better or worse depending on the oracle's quality.*

For a given heuristic search algorithm, the guiding effect of heuristic values and the effect of the oracle are composable.

## B.2 MODELING AND ANALYSIS OF INTERNAL COUPLING FILTERING STRUCTURE

This subsection provides a mathematical modeling and analysis of the internal coupling system within the NNFC framework. The system employs a series coupling structure of neural networks to enhance filtering sensitivity and protect the Chain-of-Logic (CoL) workflow. All parameters are shown to depend on the neural network's error rate $\epsilon$.

ERROR TRANSFER AND AMPLIFICATION

The output error of each neural network in the series depends on its own parameters and the input error from the previous level, modeled through a first-order Taylor approximation, and the cumulative error amplifies through the series, depending on the error transfer coefficient $\gamma$ and base error rate $\epsilon$.

*Proof.* For the $i$-th neural network, the ideal output is $f_i(\theta_i)$, where $\theta_i$ represents the network parameters. The input includes the error from the previous level $e_{i-1}$, and the actual output is influenced by a scaling factor $k_i \in [0,1]$ and an error term $\epsilon_i$ with expectation $\epsilon$. The output is expressed as:

$$\pi_i = f_i(\theta_i + k_i \cdot e_{i-1}) + \epsilon_i$$

Using a first-order Taylor expansion around $\theta_i$:

$$\pi_i \approx f_i(\theta_i) + \frac{\partial f_i}{\partial \theta_i} \cdot k_i \cdot e_{i-1} + \epsilon_i$$

Define the error transfer coefficient $\gamma_i = \frac{\partial f_i}{\partial \theta_i} \cdot k_i$, which captures the network's ability to handle input error. The single-level error is then:

$$e_i = \pi_i - f_i(\theta_i) \approx \gamma_i \cdot e_{i-1} + \epsilon_i$$

Here, $\gamma_i < 1$ indicates error attenuation, $\gamma_i > 1$ indicates error amplification, and $\gamma_i$ is typically treated as a constant for simplicity.

Now, consider the cumulative error in a series of $K$ neural networks. Assume the error from the initial level is zero ($e_0 = 0$). The error at each level is derived step by step:

Level 1:

$$e_1 = \epsilon_1$$

Level 2:

$$e_2 \approx \gamma_2 \cdot e_1 + \epsilon_2$$

Generally, for level $k \geq 2$:

$$e_k \approx \sum_{j=1}^{k} \left( \prod_{i=j+1}^{k} \gamma_i \right) \epsilon_j$$

where $\prod_{i=j+1}^{k} \gamma_i$ is defined as 1 for $j = k$. Assuming identical error rates $\epsilon_i = \epsilon$ and error transfer coefficients $\gamma_i = \gamma$ for all networks:

$$e_k \approx \epsilon \sum_{j=0}^{k-1} \gamma^j = \epsilon \frac{1 - \gamma^k}{1 - \gamma} \quad \text{for } \gamma \neq 1$$

If $\gamma = 1$, the error simplifies to:

$$e_k \approx k\epsilon$$

This shows that error amplifies with increasing $k$. $\qquad\square$

ANALYSIS OF SERIES ENHANCEMENT ON FILTERING EFFECTIVENESS

The series coupling structure enhances filtering effectiveness by increasing the inconsistency indicator $D$ compared to a parallel structure, particularly in low-error regimes, due to error accumulation. This enhancement is analyzed through numerator-dominated and denominator-dominated cases.

*Proof.* The filtering effectiveness is assessed by comparing the inconsistency indicator $D$ for series coupling with that of a parallel structure (no series), denoted $D_{\text{avg}}$. For the parallel structure, the output is the average $\pi_{\text{avg}} = \frac{1}{K} \sum \pi_i$, and the inconsistency indicator is:

$$D_{\text{avg}} = \frac{\max_{i,j} \|\pi_i - \pi_j\|}{\|\pi_{\text{avg}}\|}$$

$D_{\text{avg}}$ serves as a baseline for comparison. - **Numerator-dominated case (low $\epsilon$):** When $\epsilon$ is small and parameters are optimized, $\|\pi_1\| \approx \|\pi_{\text{avg}}\| \approx \|\pi^*\|$, so $D$ and $D_{\text{avg}}$ are determined by their numerators. The expected numerator for series coupling, $E[R_{\text{serial}}]$, is at least $\max_i E[\|\pi_i - \pi_1\|]$. Since the output variance $\sigma_i^2 \geq \sigma^2$ (base variance),

$$E[\|\pi_i - \pi_1\|] \geq \sqrt{\sigma_i^2 + \sigma^2} \geq \sqrt{2}\sigma$$

For the parallel structure, $E[R_{\text{parallel}}] \propto \sqrt{\epsilon}$ based on extreme value theory. Thus,

$$D \propto \frac{E[R_{\text{serial}}]}{\|\pi^*\|} > \frac{E[R_{\text{parallel}}]}{\|\pi^*\|} \propto D_{\text{avg}}$$

This indicates that series coupling has a larger $D$, leading to higher filtering sensitivity. - **Denominator-dominated case (high $\epsilon$):** When $\epsilon$ is large, $\|\pi_1\|$ may increase due to positive bias, while $\|\pi_{\text{avg}}\|$ remains closer to $\|\pi^*\|$ (smaller). If the numerators are similar,

$$D = \frac{\max \|\pi_i - \pi_j\|}{\|\pi_1\|} < \frac{\max \|\pi_i - \pi_j\|}{\|\pi_{\text{avg}}\|} = D_{\text{avg}}$$

This could reduce filtering sensitivity, but parameter optimization (e.g., increasing series count $K$ or adjusting error transfer coefficient $\gamma$) can mitigate this issue globally. Series coupling generally maintains enhancement over the parallel baseline. $\qquad\square$

FILTER PROBABILITY ANALYSIS

All key parameters related to filtering probability, including the inconsistency indicator, threshold, series count, and filtering probability itself, depend on the neural network's error rate $\epsilon$. This dependency ensures that the filtering mechanism adapts to error conditions.

*Proof.* From empirical observations and the error model, the error transfer coefficient relates to $\epsilon$ as:

$$\gamma \propto 1 - \epsilon$$

This is because higher error rates increase network uncertainty, . The inconsistency indicator $D$ is defined as:

$$D = \frac{\max_{i,j} \|\pi_i - \pi_j\|}{\|\pi_1\|}$$

where $\|\pi_1\|$ is used due to its minimal error ($\epsilon$). Since $\|\pi_i - \pi_j\| \propto e_k$ and $e_k \propto \epsilon$ from Theory 1, it follows that:

$$D \propto e_{\text{coupled}}^{(K)} \approx \epsilon \frac{1 - \gamma^K}{1 - \gamma}$$

to minimize error accumulation. The filtering probability $p_{\text{filter}}$ is directly proportional to $\epsilon$:

$$p_{\text{filter}} = P(D > T) \approx \beta \cdot D \propto \beta \cdot \epsilon \cdot \frac{1 - \gamma^K}{1 - \gamma}$$

where K is the series count, T is threshold, and $\beta$ is a proportionality coefficient. $\square$

In conclusion, series coupling outperforms parallel structures in most scenarios, especially when optimized.

## B.3    Dynamic Error Analysis

### Mathematical Modeling of the Neural Network Error

In the NNFC framework, the neural network module is modeled as a parametric function that outputs a rule selection strategy $\pi(a|s; \theta)$, where $s$ is the state (e.g., current partial program), $a$ is the action (e.g., rule application), and $\theta$ is the network parameter. The performance of the neural network is evaluated through error terms. The total error $e_{\text{total}}$ consists of a static error $e$ and a dynamic error $e_{\text{dynamic}}$:

$$e_{\text{total}} = e + e_{\text{dynamic}}$$

where the dynamic error $e_{\text{dynamic}}$ is the sum of drift error $e_{\text{drift}}$ and forget error $e_{\text{forget}}$:

$$e_{\text{dynamic}} = e_{\text{drift}} + e_{\text{forget}}$$

**Definition B.3** (Static Error $e$). The static error $e$ arises from the neural network's limited expressiveness or insufficient training data, representing the residual error after convergence, with $e \geq 0$. It quantifies the difference between the strategy $\pi_{n-1}$ and the ideal strategy $\pi_{n-1}^*$:

$$e = L(\pi_{n-1}, \pi_{n-1}^*)$$

where $L$ is a loss function (e.g., cross-entropy). Under ideal conditions (e.g., convex loss function and adaptive learning rate), training converges to $e = 0$; however, due to expressiveness constraints, $e$ typically converges to a non-zero value.

**Definition B.4** (Drift Error $e_{\text{drift}}$). The drift error $e_{\text{drift}}$ results from environmental changes (e.g., DSL rule modifications or task distribution shifts), causing input distribution drift. It is modeled as proportional to the rate of change of the ideal strategy:

$$e_{\text{drift}} = \alpha \|\pi_n^* - \pi_{n-1}^*\|$$

where $\alpha$ is a coefficient indicating sensitivity to drift. This error accumulates over time if unresolved.

**Definition B.5** (Forget Error $e_{\text{forget}}$). The forget error $e_{\text{forget}}$ occurs during continual learning due to performance degradation from new data overwriting old knowledge. It depends on the strategy change rate and neural network parameters (e.g., learning rate $\eta$ and network capacity $C$):

$$e_{\text{forget}} = \gamma \sum_{i=1}^{n-1} \|\pi_i - \pi_{i-1}\|$$

where $\gamma$ is a composite coefficient with $\gamma \propto \eta/C$. Higher $\eta$ or lower $C$ increases $\gamma$, exacerbating forgetting.

The neural network agent is thus modeled as a tuple: output strategy $\pi$, training error $e$, and drift error $e_{\text{drift}}$. Additionally, the module includes an inner coupling consistency check mechanism to filter erroneous predictions. Filtered strategies are denoted $\pi_{\text{none}}$. The filtering effect is controlled by two parameters: the number of serial neural networks $K$ and a consistency threshold $\tau$. The filtering probability $p_{\text{filter}}$ is proportional to $K \cdot e_{\text{total}}$, meaning higher error or more serial networks enhance filtering. Inner coupling ensures that filtering for erroneous predictions is significantly more effective than for correct ones.

LYAPUNOV ANALYSIS

To analyze system stability, we define an energy function $V$ as the sum of two components:

$$V = V_{\text{neural}} + V_{\text{output}}$$

where: - $V_{\text{neural}} = \|\pi^* - \pi_{\text{neural}}\|^2$ is the neural network error term, measuring the deviation between the ideal strategy $\pi^*$ and the neural network output strategy $\pi_{\text{neural}}$. - $V_{\text{output}} = \|\pi^* - \pi_{\text{output}}\|^2$ is the system output error term, measuring the deviation between the ideal strategy and the final output strategy $\pi_{\text{output}}$.

The energy change $\Delta V = V_n - V_{n-1}$ determines stability. We analyze each component separately.

ANALYSIS OF $V_{\text{NEURAL}}$

The change in $V_{\text{neural}}$ is influenced by the training process and dynamic error. Due to gradient-based learning, $\|\pi^* - \pi_{\text{neural}}\|^2$ typically decreases, but dynamic error may increase it. The change can be expressed as:

$$\Delta V_{\text{neural}} \leq -\eta \|\nabla L(\pi_{\text{neural}})\|^2 + \delta \cdot e_{\text{dynamic}} + O(\eta^2)$$

where: - $\eta$ is the learning rate, - $\|\nabla L(\pi_{\text{neural}})\|^2$ is the gradient norm of the loss function, - $\delta$ is a coefficient, - $e_{\text{dynamic}} = e_{\text{drift}} + e_{\text{forget}}$ is the dynamic error (sum of drift and forget errors), - $O(\eta^2)$ represents higher-order terms, negligible for small $\eta$.

This inequality holds when the loss function $L$ is smooth and convex, ensuring that gradient descent reduces error in expectation. The dynamic error $e_{\text{dynamic}}$ acts as a disturbance, potentially hindering convergence.

ANALYSIS OF $V_{\text{OUTPUT}}$

The change in $V_{\text{output}}$ is derived based on the dynamics of the output error $e_{\text{output}} = \|\pi^* - \pi_{\text{output}}\|$. The output error change $\Delta e_{\text{output}}$ is influenced by learning and dynamic error injection:

$$\Delta e_{\text{output}} \approx -\alpha \rho e_{\text{output}} + \beta e_{\text{dynamic}}$$

where: $\alpha > 0$ is the learning efficiency coefficient, $\rho = \kappa(1 - e_{\text{output}})$ is the termination rate with sensitivity coefficient $\kappa > 0$, $\beta > 0$ is the dynamic error injection coefficient, which scales the different source dynamic errors as a single source, simplifying the model by combining input and neural processing errors into a unified term. $e_{\text{dynamic}}$ is the dynamic error, comprising drift and forget errors. It originates from both the input dynamic error and the neural network's processing dynamic error during symbolic structure manipulation. However, the neural processing component dominates, so $e_{\text{dynamic}}$ is effectively treated as a scaled version of this dominant error, ensuring a simplified analysis.

The energy change for $V_{\text{output}}$ is then:

$$\Delta V_{\text{output}} \approx 2e_{\text{output}}\Delta e_{\text{output}} = 2e_{\text{output}}(-\alpha \rho e_{\text{output}} + \beta e_{\text{dynamic}})$$

Substituting $\rho = \kappa(1 - e_{\text{output}})$:

$$\Delta V_{\text{output}} = -2\alpha\kappa(1 - e_{\text{output}})e_{\text{output}}^2 + 2\beta e_{\text{dynamic}}e_{\text{output}}$$

For stability of $V_{\text{output}}$, we require $\Delta V_{\text{output}} \leq 0$, which implies:

$$-2\alpha\kappa(1 - e_{\text{output}})e_{\text{output}}^2 + 2\beta e_{\text{dynamic}}e_{\text{output}} \leq 0$$

Assuming $e_{\text{output}} > 0$, divide both sides by $2e_{\text{output}}$:

$$-\alpha\kappa(1 - e_{\text{output}})e_{\text{output}} + \beta e_{\text{dynamic}} \leq 0$$

Rearranging:

$$\beta e_{\text{dynamic}} \leq \alpha\kappa(1 - e_{\text{output}})e_{\text{output}}$$

The right-hand side is a quadratic function $f(e_{\text{output}}) = \alpha\kappa(1 - e_{\text{output}})e_{\text{output}}$, which achieves its maximum at $e_{\text{output}} = \frac{1}{2}$ with value $\frac{\alpha\kappa}{4}$. Thus, the stability condition is:

$$e_{\text{dynamic}} \leq \frac{\alpha\kappa}{4\beta}$$

This means that for $V_{\text{output}}$ to be stable, the dynamic error must not exceed the critical value $\frac{\alpha\kappa}{4\beta}$.

OVERALL ENERGY CHANGE AND STABILITY

The overall energy change is the sum of the components:

$$\Delta V = \Delta V_{\text{neural}} + \Delta V_{\text{output}}$$

Stability requires $\Delta V \leq 0$, which depends on:

**Training effectiveness:** A large gradient norm $\|\nabla L\|$ improves convergence, as it indicates significant progress in reducing error. This is typically ensured when the loss function satisfies the Polyak-Łojasiewicz (PL) condition or is strongly convex.

**Dynamic error size:** A small $e_{\text{dynamic}}$ reduces the negative impact on stability. Specifically, when $e_{\text{dynamic}} = 0$, the system converges to a local minimum; when $e_{\text{dynamic}}$ is small, the system remains stable if training effectiveness dominates.

**Filtering mechanism:** When filtering is inactive ($p_{\text{filter}} = 0$), stability is sensitive to $e_{\text{dynamic}}$; when active ($p_{\text{filter}} > 0$), filtering reduces errors but may introduce delay.

In practice, the system may not converge to zero error due to $e_{\text{dynamic}}$, but it can reach a bounded steady-state error $e_{\text{steady}} \approx \frac{\beta e_{\text{dynamic}}}{\alpha \kappa}$ when $e_{\text{dynamic}}$ is small. Meanwhile, termination rate $\rho$ is negatively correlated with the output error $e_{\text{output}}$ (since $\rho = \kappa(1 - e_{\text{output}})$), which is influenced by the neural error $e_{\text{neural}}$. Therefore, filtering is important to ensure a minimal value of $\rho$ ($\rho_{\min} > 0$, ), which maintains learning opportunities and prevents divergence when the output error is high."

STABILITY CONCLUSION

The Lyapunov analysis shows that system stability depends on the balance between training effectiveness and dynamic error. The energy function $V = V_{\text{neural}} + V_{\text{output}}$ provides a framework for assessing convergence. By ensuring $\Delta V \leq 0$ through controlled dynamic error and effective learning, the system can maintain stability. Filtering plays a crucial role in handling high error scenarios, preventing divergence, and promoting recovery. This analysis guides parameter selection and system design in the NNFC framework.

## C  COOL HOST LANGUAGE

COOL employs a hybrid reasoning approach with dynamic typing and support for numerical operations, enabling flexible DSL management. The framework allows random access storage and read/write operations, providing the expressiveness required for coordinating heterogeneous DSLs. A COOL rule example is shown in Figure 5, demonstrating the integration of the heuristic vector and keywords.

### C.1  SPECIFICATION

**Code C.1: Relational Reasoning Task Input Program**

```
(Wesley) is (James)s son & (Martha) is (Wesley)s daughter &
(Hugh) is (Martha)s uncle & (Hugh) is (James)s ($relation);
```

where $ specifies the nonterminal, indicating that the COOL framework coordinates multiple DSLs to compute the correct value for `relation` (the relationship between `Hugh` and `James`) through dynamic regulation. COOL does not explicitly assign specifications to individual DSLs; instead, it enables collaborative reasoning across DSLs, where each DSL actively identifies and solves applicable specifications based on its domain logic, facilitated by CoL's keywords and NNFC's adaptive control.

COOL employs a unified compilation approach, where reasoning occurs at the intermediate representation level (see Appendix N). Codes **?? ??** are provided for illustrative purposes to demonstrate the multi-DSL regulation process facilitated by CoL's keywords and NNFC's adaptive control.

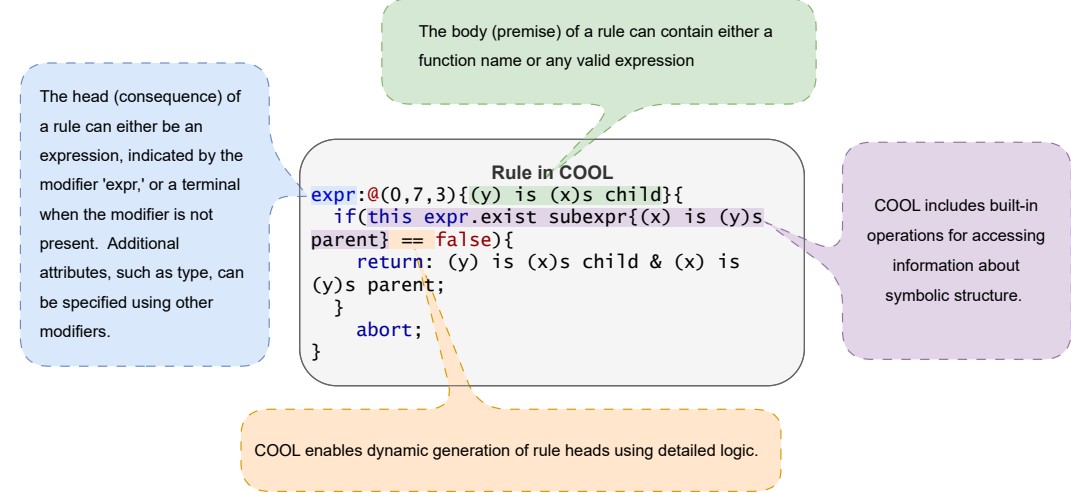

Figure 5: DSL rules in COOL. The framework allows for defining rule heads using expressions or terminals, which are enhanced with modifiers for additional attributes. Rule bodies can incorporate any valid expression or function name. Besides, COOL provides built-in operations for accessing symbolic structure information and facilitates dynamic rule head generation.

## C.2  TRADITIONAL DSL

This appendix presents a typical DSL model to facilitate understanding of related concepts and experimental metrics in the paper. A typical DSL is defined as a context-free grammar:

$$G = \{V, \Sigma, R, S\}, \tag{1}$$

where $V$ is the set of non-terminal symbols, $\Sigma$ is the set of terminal symbols, $R$ is the set of rules, and $S$ is the starting symbol. The DSL's derivation process converts intermediate states containing non-terminal into complete outputs by applying given rules.

The derivation process for traditional DSLs involves iteratively transforming partial programs into complete programs by applying a series of rules. Each intermediate state $(s)$ and the corresponding rule $(r)$ applied form a transformation pair $(s, r)$. Rule applications modify the symbolic structures via tree operations, and a sequence of these operations constitutes a derivation trajectory. These trajectories are classified into three types:

- **Feasible Path**: Leads to a complete solution.

- **Infeasible Path**: Cannot yield a valid solution.

- **Unterminated Path**: Process still in progress.

To clarify key concepts involved in the synthesis process, we provide the following definitions of terms:

- **Tree Operation/Manipulation**: Refers to the modification of the syntax tree of a during the derivation process and has an associated CPU cost.

- **Transformation Pair** $(s, r)$: A combination of a intermediate state and a rule to be applied. It records the explored space and possible exploration directions, requiring memory storage.

- **Derivation Path/Trajectory**: A sequence of transformation pairs, $\{(s_0, r_0), (s_1, r_1), \dots\}$, representing the process of transforming an initial state with non-terminal symbol into a complete one. Its function is to track the entire derivation process, whether it leads to a feasible, infeasible, or Unterminated path.

## C.3 A* Search in Program Synthesis

During the exploration phase of reasoning process, we leverage the A* algorithm to perform the heuristic search. This algorithm is renowned for its efficacy in discrete optimization tasks, utilizing heuristic guidance to navigate the search space effectively Hart et al. (1968). Each action or decision is associated with a specific cost in this context. By evaluating the cumulative cost of actions taken so far and the estimated costs of future actions, A* seeks to determine the path with the least overall cost. In COOL, heuristic values promoting forward progression are considered rewards. Therefore, we treat them as negative costs in calculations. Algorithm 1 illustrates the implementation details.

---

**Algorithm 1** Search Algorithm in COOL

---

  **function** A* Search $(initialState, u_2)$
    $openSet \leftarrow$ priority queue containing only the initial state
    $gScore[startState] \leftarrow 0$ {cost from start}
    $fScore[startState] \leftarrow 0$
    **while** $openSet \neq \emptyset$ **do**
      $currentState \leftarrow openSet.\text{pop}(\ )$ {State in openSet with lowest fScore value}
      **if** $currentState$ is complete output **then**
        **return** Success
      **end if**
      **for all** $neighbor \in$ neighbors of $currentState$ **do**
        { Neighbor is obtained by applying a rule to the current state}
        $tentative\_gScore \leftarrow gScore[current] - u_2[neighbor]$
        **if** $tentative\_gScore < gScore[neighbor]$ **then**
          $cameFrom[neighbor] \leftarrow current$
          $gScore[neighbor] \leftarrow tentative\_gScore$
          $fScore[neighbor] \leftarrow gScore[neighbor] - u_2[neighbor]$
          **if** $neighbor \notin openSet$ **then**
            $openSet.\text{add}(neighbor)$
          **end if**
        **end if**
      **end for**
    **end while**
    **return** Failure
  **end function**

---

## C.4 CoL Coordination Process

The reasoning process in CoL involves coordinated operation across multiple DSLs, each handling specific aspects of the regulation task. These DSLs operate sequentially, progressively refining symbolic representations through structured transformations. The key distinction from common DSLs is that intermediate DSLs can generate partial outputs that serve as inputs for subsequent DSLs in the coordination chain.

Each DSL in the CoL framework focuses on a specific reasoning aspect to incrementally transform the symbolic representation. For instance, in Figure 2:

- The first DSL separates relations and genders, decomposing the initial representation into simpler components for processing

- The second DSL reasons about inverse relationships, further structuring the intermediate representation by identifying inverse connections

- The third DSL handles indirect relationships, providing additional contextual information to relationships identified earlier

- The final DSL recombines relations and genders while eliminating irrelevant relations to produce a complete solution

During each DSL's operation, the reasoning process utilizes heuristic vectors to prioritize rule application, focusing on areas most likely to lead to successful coordination outcomes. For intermediate

DSLs, where completion cannot be determined based solely on final outputs, guidance through heuristic values and coordination flow control using keywords becomes essential for effective multi-DSL regulation.

## D NEURAL NETWORKS IN COOL

COOL incorporates an integrated machine learning system that automatically collects data generated during multi-DSL regulation and performs training and prediction tasks for neural networks within the framework.

### D.1 DATA COLLECTION AND COMBINATION FOR TRAINING

The neural networks utilize transformation pairs $(s, r)$ from derivation trajectories to train various heads for multi-DSL coordination.

To train the neural networks for regulation tasks of type $T$, COOL constructs the dataset as follows:

**Task Detection Head (TDH)**: This head determines whether the input intermediate state belongs to task type $T$. This is a binary classification task. Intermediate states from type $T$ derivation trajectories are collected as positive examples (proportion: 67%), while states from other derivation trajectories and built-in function calls are collected as negative examples (proportion: 33%).

**Search Space Prune Head (SSPH)**: After confirming the state belongs to type $T$, this head identifies whether the input intermediate state can be feasibly derived into a complete solution. This is also a binary classification task. States from feasible derivation trajectories are collected as positive examples (proportion: 67%), while states from infeasible trajectories are collected as negative examples (proportion: 33%).

**Search Guidance Head (SGH)**: After determining that the input is a feasible type $T$ intermediate state, this head generates rule features to guide the DSL coordination process, involving classification and regression tasks.

### D.2 NEURAL NETWORK INPUT

As shown in Figure 3, there are three neural networks in a neural agent. Each network (labeled A, B, and C in their sequential order) takes an intermediate symbolic structure as input. The intermediate symbolic structure is represented at the intermediate representation (IR) level in the form of Three-Address Code (TAC) (see Appendix N), allowing derivation process to be conducted without the constraints of specific DSL syntax or the machine code format of the execution platform Sujeeth et al. (2014). The TAC is then transformed into a graph representation for input to neural networks.

In the serial coupling structure of the neural agent, network B is the downstream neural network of A and uses the output of the SGH head from A as part of its input. Similarly, network C is the downstream neural network of B and uses the output of the SGH head from B as part of its input. This serial coupling enables each downstream network to accumulate the error produced by the upstream network, making incorrect predictions more obvious.

The specific input features are shown in Table 4.

### D.3 NEURAL NETWORK ARCHITECTURE

As TAC embodies both the graphical properties of a symbolic structure and the sequential properties of derivation, the design of the neural network must be capable of capturing these dual characteristics.

The detailed layer architecture of neural networks in COOL is illustrated in Figure 6. The processing flow consists of the following steps:

1. **Embedding Node Features:** We start by employing embedding layers with learning capabilities. These layers convert categorical inputs into dense, continuous vectors, which enhances the stability and efficiency of subsequent processing layers Hrinchuk et al. (2019).

Table 4: Input features of neural networks in COOL. Each entry specifies the feature, its size, and the neural networks it pertains to, along with a description of its role. These features contribute to the neural network's understanding of the symbolic structure's state and semantics, aiding in accurate multi-DSL regulation.

| Feature | Feature Size | Neural Network | Signification |
|---|---|---|---|
| grounded | 2 | A, B, C | The node is in a fully specified expression. |
| domain | 1 | A, B, C | Domain of the subtask represented by the subtree where the node is located. |
| root | 2 | A, B, C | The tree representing the subtask is rooted at this node. |
| non-terminal | 2 | A, B, C | The node is a non-terminal. |
| type | 1 | A, B, C | Type of the node. |
| identifier | 1 | A, B, C | Identifier of the node. |
| string | 1 | A, B, C | The node contains a string as the immediate value. |
| number | 1 | A, B, C | The node contains a number as the immediate value. |
| operator | 1 | A, B, C | The node is an operator. |
| current stage | 1 | A, B, C | Current CoL stage (valid when this node is grounded). |
| operand position | 3 | A, B, C | Placement of nodes in a binary operation tree (left operand node, right operand node, operation node). |
| applied (SGH) | 1 | B, C | A rule is applied to the subtree rooted at this node (derived from the output feature "**jumps**" of the previous neural network). |
| next stage (SGH) | 1 | C | The CoL stage to advance to after applying the rule (derived from the output feature "**next stage**" of the previous neural network). |

Table 5: Output features of neural networks in DSNN. These features provide comprehensive optimizations for CoL DSL during program synthesis, including task detection, search space pruning, and search guidance.

| Feature | Feature Size | Neural Network | Signification |
|---|---|---|---|
| domain (TDH) | 2 | A, B, C | Relevance of task domains to DSNN. |
| feasibility (SSPH) | 2 | A, B, C | Feasibility of generate the complete output. |
| jumps (SGH) | max_tree_depth*3 | A, B, C | The path from the tree's root to the subtree's root where the rule is applied (jump left, right, or stop in each step). |
| next stage (SGH) | 1 | A, B, C | The CoL stage to advance to after applying the rule. |
| heuristic sign (SGH) | 2 | A, B, C | Sign of the rule's heuristic value. |
| heuristic value (SGH) | 1 | A, B, C | Rule's heuristic value. |
| expression (SGH) | 2 | A, B, C | Type of rule's head (expression or terminal). |

2. **Graph Feature Extraction:** Next, we use a Graph Neural Network (GNN) to extract graph features from each line of TAC representation Drori et al. (2022); Wu et al. (2022). To adaptively extract intricate details such as node types, graph attention (GAT) layers are applied after the embedding layers Velickovic et al. (2017).

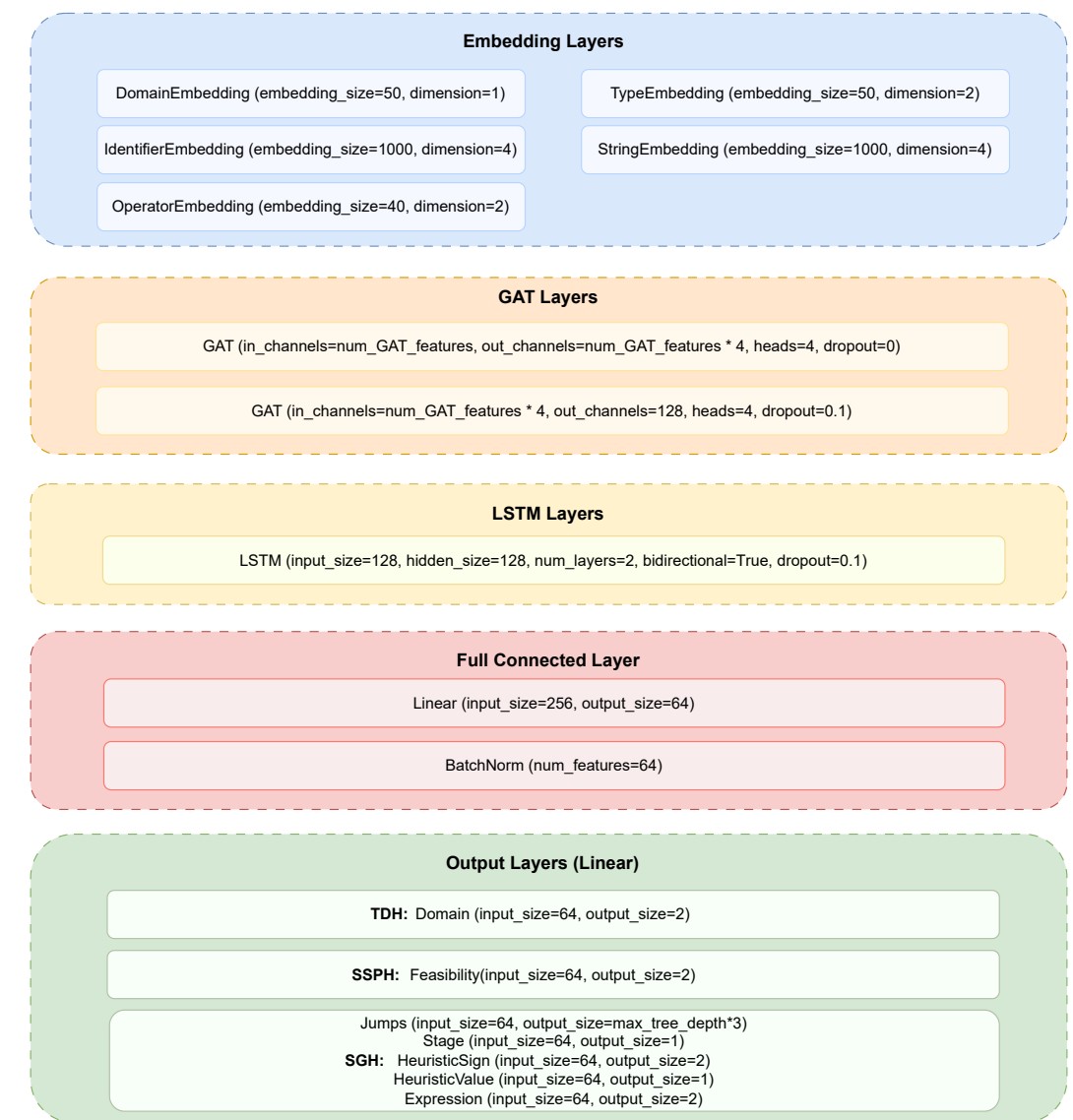

Figure 6: Layer architecture of neural networks in COOL. Each neural network consists of embedding layers for domains, types, identifiers, strings, and operators, followed by GAT layers for structural feature extraction. LSTM layers provide sequential modeling for derivation trajectories, with fully connected layers combining the outputs. Various output layers handle domain identification for task detection, feasibility judgment for search space pruning, state transitions, stage prediction, heuristic constraint (sign and value), and constraint on the type of rule's head (expression or terminal) for search guidance.

3. **Sequential Feature Processing:** We adopt Long Short-Term Memory (LSTM) networks to capture the sequential features inherent in TAC Chen et al. (2021); Nye et al. (2020). Recognizing the equal importance of each TAC line, bidirectional LSTM layers are employed following the GAT layers to enrich the contextual understanding Huang et al. (2015).

4. **Multi-Head Output:** Finally, the processed data is channeled through multiple output layers to prevent task interference and ensure clarity in results.

Figure 3 (right) illustrates the use of three neural network units arranged in series to construct the internal coupling structure of the NNFC mechanism. Labeling these neural networks as A, B, and C

in sequential order, Table 4 details the specific input features for each network: Neural network B receives its input feature "applied" from network A's output feature "jumps," while network C's input features "applied" and "next stage" are derived from the output features "jumps" and "next stage" of network B. The output features of the three neural network units are consistent and comparable. Table 5 presents the output features of these networks.

### D.4 PREDICTION FILTERING

By comparing the output differences of the heads, we can determine whether there are possible prediction errors and filter out the prediction results. For classification tasks, we directly compare whether the outputs are the same. For regression tasks, we set a tolerance threshold (10%) for the difference.

### D.5 PREDICTION ACTING ON HEURISTIC VALUES

As shown in Figure 1, the heuristic value of a rule affects its application, and the prediction results of the neural networks affect the derivation process by correcting the heuristic value of the rules in the sub-DSL based on the output of heads:

**Task Detection Head (TDH)**: If the output indicates that the intermediate state does not belong to the derivation task that the DSL can handle, any rule application on this state will receive an additional negative bonus on the heuristic value. For example, $\mathbf{h}[i] = \mathbf{h}[i] - |\mathbf{h}[i]| - 10$.

**Search Space Prune Head (SSPH)**: If the TDH output indicates that the intermediate state falls within the DSL and the SSPH output considers the state infeasible, any rule application on this state will receive an additional negative bonus on the heuristic value. For example, $\mathbf{h}[i] = \mathbf{h}[i] - |\mathbf{h}[i]| - 10$.

**Search Guidance Head (SGH)**: If the TDH output indicates that the intermediate state falls within the DSL and the SSPH output indicates that the state is promising for derivation into a complete solution, then if the features of the output rule match certain rules (logical values must be equal, and numerical values must fall within a $\pm10\%$ range), the heuristic value when applying these rules will receive a positive bonus. For example, $\mathbf{h}[i] = \mathbf{h}[i] + |\mathbf{h}[i]|$. Otherwise, it will receive a negative bonus: $\mathbf{h}[i] = \mathbf{h}[i] - |\mathbf{h}[i]| - 10$.

## E SIGNAL CLIPPER

The Clipper, as illustrated in Figure 3 (left), modulates coordination signals between multiple CoL instances by capping signals that do not align with neural guidance to zero:

$$u_2 = \begin{cases} 0 & \text{if } u_1 > 0 \text{ and the current CoL rule doesn't align with} \\ & \quad \text{the agent guidance, while there exists another CoL instance} \\ & \quad \text{whose rule aligns with the guidance within the search space} \\ u_1 & \text{otherwise} \end{cases} \quad (2)$$

This mechanism regulates signal interactions between CoL instances to suppress unnecessary interleaving during multi-DSL coordination.

## F EXPERIMENT

### F.1 TASK EXAMPLE

The benchmarks in the experiment are COOL code that representing the specifications.

#### F.1.1 RELATIONAL TASKS

Used to test the performance of multi-DSL regulation of a single task type, completeness guaranteed:

### Code F.1: Relational Task Example

```
//load 4 DSLs for family relationship reasoning
#load(family)

//Relational reasoning questions like (50 per batch):
(Wesley) is (James)s son & (Martha) is (Wesley)s daughter &
(Hugh) is (Martha)s uncle & (Hugh) is (James)s ($relation);
...
```

#### F.1.2 SYMBOLIC TASKS

Used to test the performance of multi-DSL regulation of a single task type, completeness not guaranteed:

### Code F.2: Symbolic Task Example

```
//load 7 DSLs for family relationship reasoning and symbolic
reasoning
#load(quadratic)

//Symbolic reasoning questions like (50 per batch):
$x^2 + 4*$x == 3;
...
```

#### F.1.3 CROSS-TYPE TASKS

Used to test the performance of multi-DSL regulation of different task types, completeness not guaranteed:

### Code F.3: Cross-Type Task Example

```
//load 11 DSLs for relational reasoning and symbolic
reasoning
#load(family)
#load(quadratic)

//Symbolic reasoning questions like (50 per batch):
$x^2 + 4*$x == 3;
...
//Relational reasoning questions like (50 per batch):
(Wesley) is (James)s son & (Martha) is (Wesley)s daughter
& (Hugh) is (Martha)s uncle & (Hugh) is (James)s ($relation);
...
```

The execution of the code that does not contain the task specifications is represented as a control variable in the experiment and is deducted from the final experimental results.

#### F.2 GROUP CONFIGURATION

The specific experimental groups of dynamic and static experiments and their respective settings are shown in Table 6:

Table 6: Group configurations. Groups marked with ★ are the main experiments, those with ☆ are for ablation and extended experiments. **Direct** indicates using DSL rules without coordination mechanisms, **Flt** indicates employing the neural filtering structure

| Group | Experiment | Trained DSNN | NNFC | Filtering Structure |
|---|---|---|---|---|
| Direct | static | | | |
| ☆Direct (Heuristic) | static | | | |
| ★CoL | static, dynamic | | | |
| ☆Direct+NN | static | ✓ | | |
| ☆Direct (Heuristic)+NN | static | ✓ | | |
| ☆ CoL +NN | static | ✓ | | |
| ☆CoL +NNFC | dynamic | | ✓ | |
| ☆Direct+NN (Flt) | static | ✓ | | ✓ |
| ☆Direct(Heuristic)+NN (Flt) | static | ✓ | | ✓ |
| ☆CoL +NN (Flt) | static | ✓ | | ✓ |
| ☆CoL +NN (Flt) | static | ✓ | | ✓ |
| ★CoL +NNFC (Flt) | dynamic | | ✓ | ✓ |

## F.3 MULTI-DSL CONFIGURATION

Table 7 shows the CoL used in the multi-DSL regulation tasks.

Table 7: Multi-DSL configurations.

| Benchmark | Rules | | | | | DSL Number |
|---|---|---|---|---|---|---|
| | Total | Production Rules | Reduction Rules | Recursive Rules | Permutation Rules | |
| relational | 40 | 36 | 2 | 16 | 0 | 4 |
| symbolic | 55 | 17 | 26 | 3 | 11 | 7 |
| cross task | 95 | 53 | 28 | 19 | 11 | 11 |

## F.4 EXPERIMENT DATA

Static experiment results are shown in Figure 7:

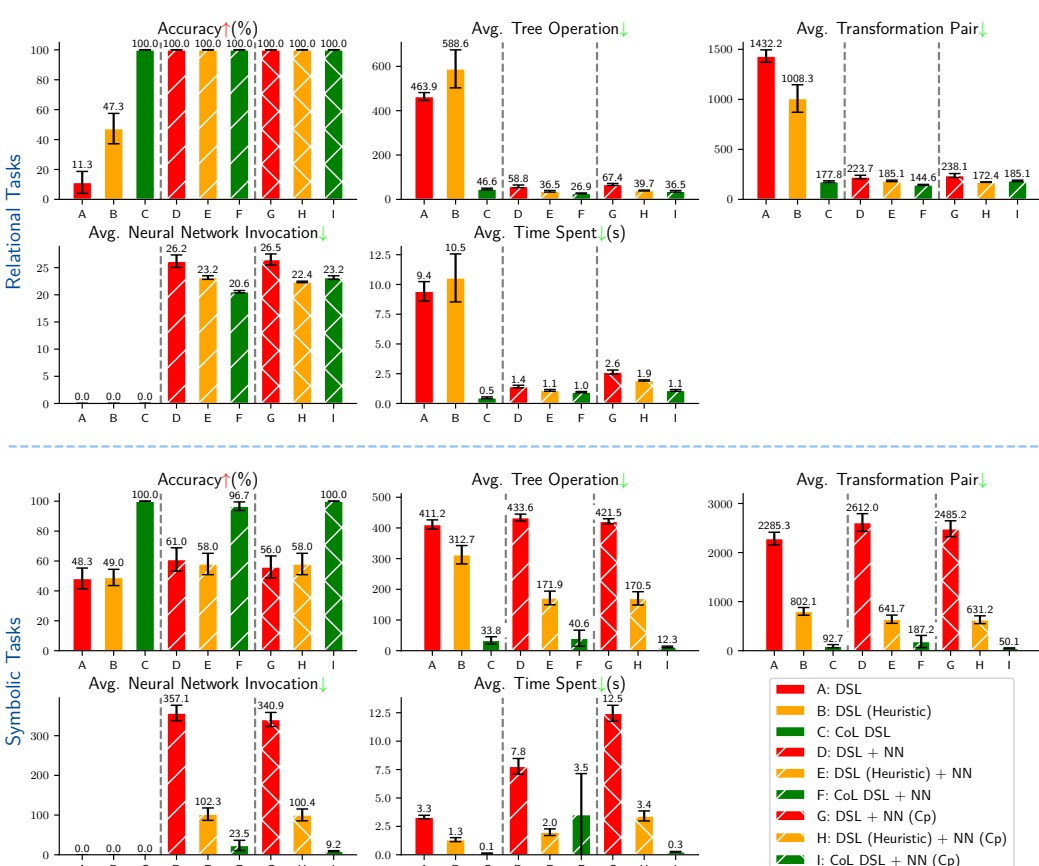

Figure 7: Static performance on relational and symbolic tasks at difficulty level A. CoL-based groups outperform Direct (Heuristic) and Direct groups. Performance varies for neural-enhanced groups with the filtering structure. Error bars show 95% confidence intervals across 6 batches.

Dynamic experiment results of single type task regulation are shown in Figure 8:

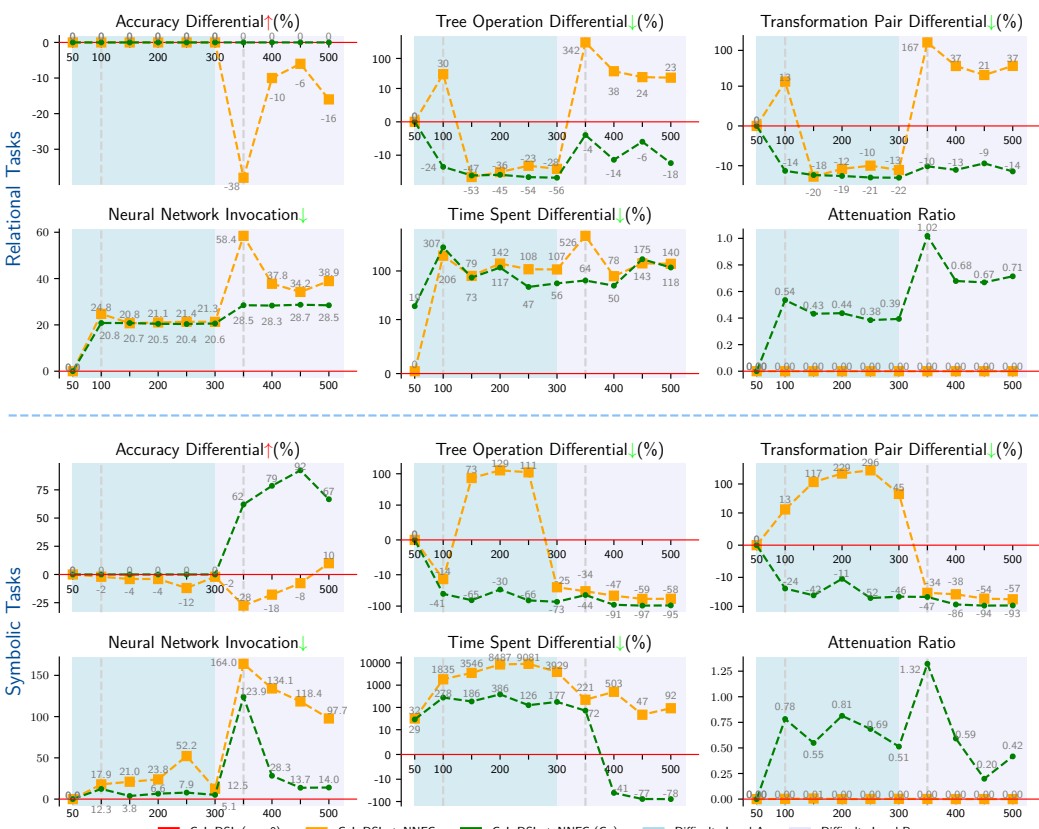

Figure 8: Dynamic performance differential to COOL in single type task coordination. The NNFC group without the filtering structure shows 12 accuracy declines across 20 batches, while the group with the structure shows none. Each batch consists of 50 derivation tasks, and NNFC continuously trains neural networks using generated regulation data after each batch, starting from scratch. (**NNFC is the neural agent tied with the CoL**)

Dynamic experiment results of cross-type task regulation are shown in Figure 9:

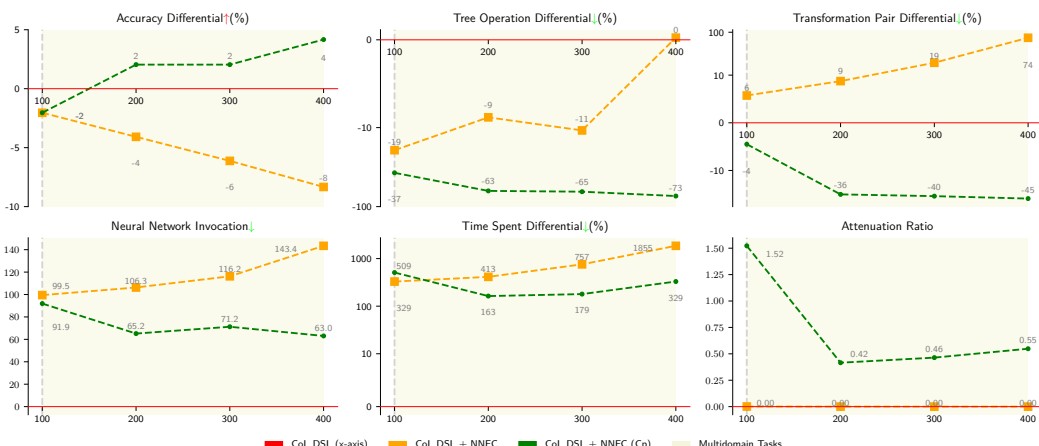

Figure 9: Dynamic performance differential to COOL in cross-type tasks coordination. The NNFC group without the filtering structure shows performance degradation across all 4 batches, while the group with the structure experiences degradation only in the first batch. Each batch includes 50 relational and 50 symbolic coordination tasks, and neural networks are continuously trained from regulation data at difficulty level A in Figure 8.

## G  IMPLEMENTATION

### G.1  IMPLEMENTATION TOOLCHAIN

To fully implement COOL's features for multi-DSL regulation, we developed the framework from the ground up using C++ as the primary language to meet the efficiency requirements of intensive tree operations during coordination processes. For syntax and semantic parsing, we employed Lex Lesk & Schmidt (1975) and YACC Johnson et al. (1975) to handle the complex rule applications across heterogeneous DSLs. The neural network components were implemented in Python using PyTorch Imambi et al. (2021) to support adaptive regulation tasks. Table 8 details the development effort across COOL's various components.

Table 8: Code Effort in COOL. Components of COOL are developed across different programming languages.

| Language | Lines | Components |
|----------|-------|------------|
| C++ | 60k | framework |
| Python | 3k | neural agent |
| Lex | 1k | syntactic parsers |
| YACC | 2k | semantic parsers |

### G.2  RELATIONAL TASK DSLs

We present the implementation COOL code for the experiments, showcasing only the core regulation framework. Please refer to the supplementary materials for complete implementation details.

```
//1 Separate Relations and Genders
expr:@(9){(a) is (b)s grandson}{
    return:(a) is male &  (a) is (b)s grandchild & (b) is (a)s
    ↪  grandparent;
}
```

```
1620   ...
1621
1622   //2 Reason Inverse Relations
1623   expr:@(0,7,3){(a) is (b)s grandchild}{
1624       if(this expr.exist subexpr{(b) is (a)s grandparent} == false){
1625           return: (a) is (b)s grandchild & (b) is (a)s grandparent;
1626       }
1627       abort;
1628   }
1629   ...
1630
1631   //3 Reason Indirect Relations
1632   expr:@(0,0,5){(a) is (b)s sibling}{
1633       placeholder:p1;
1634       while(this expr.find subexpr{(p1) is (a)s sibling}){
1635           if(this expr.exist subexpr{(p1) is (b)s sibling} == false
1636           ↪  && p1 != b){
1637   return: (a) is (b)s sibling & (p1) is (b)s sibling;
1638           }
1639           p1.reset();
1640       }
1641       p1.reset();
1642       while(this expr.find subexpr{(p1) is (a)s parent}){
1643           if(this expr.exist subexpr{(p1) is (b)s parent} == false){
1644   return: (a) is (b)s sibling & (p1) is (b)s parent;
1645           }
1646           p1.reset();
1647       }
1648       p1.reset();
1649       while(this expr.find subexpr{(p1) is (a)s pibling}){
1650           if(this expr.exist subexpr{(p1) is (b)s pibling} ==
1651           ↪  false){
1652   return: (a) is (b)s sibling & (p1) is (b)s pibling;
1653           }
1654           p1.reset();
1655       }
1656       p1.reset();
1657       while(this expr.find subexpr{(p1) is (a)s grandparent}){
1658           if(this expr.exist subexpr{(p1) is (b)s grandparent} ==
1659           ↪  false){
1660   return: (a) is (b)s sibling & (p1) is (b)s grandparent;
1661           }
1662           p1.reset();
1663       }
1664       p1.reset();
1665       abort;
1666   }
1667   ...
1668
1669   //4 Recombine Relations and Genders, Eliminate Irrelevant
1670   ↪  Relations
1671   expr:@(0,0,0,8){(a) is (b)s ($relation)}{
1672       //immediate family
1673       placeholder:p1;
       while(this expr.find subexpr{(a) is (b)s grandchild}){
           if(this expr. exist subexpr{(a) is male}){
   return: $relation == "grandson";
           }
           if(this expr.exist subexpr{(a) is female}){
   return:$relation == "granddaughter";
```

```
1674            }
1675          p1.reset();
1676        }
1677      p1.reset();
1678      while(this expr.find subexpr{(a) is (b)s child}){
1679          if(this expr. exist subexpr{(a) is male}){
1680   return: $relation == "son";
1681          }
1682          if(this expr.exist subexpr{(a) is female}){
1683   return:$relation == "daughter";
1684          }
1685          p1.reset();
1686        }
1687      ...
1688      abort;
1689  }
1690  ...
       expr:@(0,0,0,10){a & ($b == c)}{
1691      return:b == c;
1692  }
1693  ...
1694
```

## H  SYMBOLIC TASK DSLS

```
// Common Transformations
expr:@(2,2,2,2,2){0+#a}{
    return:a;
}
expr:@(2,2,2,2,2){#a+0}{
    return:a;
}
...

// 1 Expand Square Terms
expr:@(5,0,0,0){(#?a + #?b)^2}{
    return:a^2+2*a*b+b^2;
}
expr:@(5,0,0,0){(#?a - #?b)^2}{
    return:a^2+(-2)*a*b+b^2;
}
expr:@(6,0,0,0){(#a*#b)^2}{
    return:a^2*b^2;
}
...

// 2 Expand Bracketed Terms
expr:@(0,4,0,0,0){#?a-(#?b+#?c)}{
    return:a-b-c;
}
expr:@(0,3.8,0,0,0){(#?b+#?c)*#?a}{
    return:b*a+c*a;
}
...

// 3 Extract Coefficients
expr:@(0,0,5,0){$x*a}{
    return:a*x;
}
expr:@(0,0,4.8,0){(immediate:a*$x)*(immediate:b*$x)}{
```

```
     new:tmp = a*b;
     return:tmp*x^2;
}
expr:@(0,0,4.6,0){$x*(a*$x)}{
     return:a*x^2;
}
...

// 4 Re-Express Negative Coefficients
expr:@(0,0,0,3.5,0){#a-$x}{
     placeholder:p1;
     placeholder:p2;
     if(x.exist subexpr{p1*p2}){
          abort;
     }
     return:a+(-1)*x;
}
expr:@(0,0,0,3.7,0){#a-immediate:b*$x}{
     new:tmp = 0 - b;
     return:a+tmp*x;
}
...

//5 Arrange Terms in Descending Order, Combine Like Terms
expr:@(0,0,0,0,3){immediate:a*$x+immediate:b*$x}{
     new:tmp = a+b;
     return:tmp*x;
}
expr:@(0,0,0,0,2.8){a1*$x+a2*$x^2}{
     return:a2*x^2+a1*x;
}
...

//6 Convert to Standard Form
expr:@(0,0,0,0,0,2.5){a*$x^2+b*x == #d}{
     return: a*$x^2+b*x + 0 == d;

}
expr:@(0,0,0,0,0,2.5){b*$x == $d}{

     if(d.exist subexpr{x^2}){
          return: 0*x^2 + b*x + 0 == d;
     }else {
          abort;
     }
}
expr:@(0,0,0,0,0,-4){$a==$b}{
     return:b==a;
}
...

//7 Apply Solution Formula
@(0,0,0,0,0,0,0,10){a*$x^2+b*x+c==0}{
     if(b^2-4*a*c<0){
          x="null";
     }
     else {
          new:x1=(-b+(b^2-4*a*c)^0.5)/(2*a);
          new:x2=(-b-(b^2-4*a*c)^0.5)/(2*a);
          x={x1,x2};
```

```
1782        }
1783    };
1784
1785
```

## I RELATIONAL TASKS AT DIFFICULTY LEVEL A (SOURCE CODE)

```
1789    #load(family) // Load the CoL DSL library for Relational Tasks
1790    new:relation = "";
1791    // [Francisco]'s brother, [Wesley], recently got elected as a
1792    ↪  senator. [Lena] was unhappy with her son, [Charles], and his
1793    ↪  grades. She enlisted a tutor to help him. [Wesley] decided to
1794    ↪  give his son [Charles], for his birthday, the latest version
1795    ↪  of Apple watch.
1796    // Ans: (Francisco) is (Lena)s brother
1797    new:Lena = "Lena";
1798    new:Charles = "Charles";
1799    new:Wesley = "Wesley";
1800    new:Francisco = "Francisco";
1801    (Charles) is (Lena)s son & (Wesley) is (Charles)s father &
1802    ↪  (Francisco) is (Wesley)s brother & (Francisco) is (Lena)s
1803    ↪  ($relation);
        relation-->"#FILE(SCREEN)";

1804    // [Clarence] woke up and said hello to his wife, [Juanita].
1805    ↪  [Lynn] went shopping with her daughter [Felicia]. [Felicia]'s
1806    ↪  sister [Juanita] was too busy to join them.
1807    // Ans: (Lynn) is (Clarence)s mother-in-law
1808    new:Clarence = "Clarence";
1809    new:Juanita = "Juanita";
1810    new:Felicia = "Felicia";
1811    new:Lynn = "Lynn";
1812    (Juanita) is (Clarence)s wife & (Felicia) is (Juanita)s sister &
1813    ↪  (Lynn) is (Felicia)s mother & (Lynn) is (Clarence)s
1814    ↪  ($relation);
        relation-->"#FILE(SCREEN)";
1815    ...
1816
1817
```

## J RELATIONAL TASKS AT DIFFICULTY LEVEL B (SOURCE CODE)

```
1820    #load(family) // Load the CoL DSL library for Relational Tasks
1821    new:relation = "";
1822    // [Antonio] was happy that his son [Bernardo] was doing well in
1823    ↪  college. [Dorothy] is a woman with a sister named [Tracy].
1824    ↪  [Dorothy] and her son [Roberto] went to the zoo and then out
1825    ↪  to dinner yesterday. [Tracy] and her son [Bernardo] had lunch
1826    ↪  together at a local Chinese restaurant.
1827    // Ans: (Roberto) is (Antonio)s nephew
1828    new:Antonio = "Antonio";
1829    new:Bernardo = "Bernardo";
1830    new:Tracy = "Tracy";
1831    new:Dorothy = "Dorothy";
1832    new:Roberto = "Roberto";
1833    (Bernardo) is (Antonio)s son & (Tracy) is (Bernardo)s mother &
1834    ↪  (Dorothy) is (Tracy)s sister & (Roberto) is (Dorothy)s son &
1835    ↪  (Roberto) is (Antonio)s ($relation);
        relation-->"#FILE(SCREEN)";
```

```
// [Bernardo] and his brother [Bobby] were rough-housing. [Tracy],
↪   [Bobby]'s mother, called from the other room and told them to
↪   play nice. [Aaron] took his brother [Bernardo] out to get
↪   drinks after a long work week. [Tracy] has a son called
↪   [Bobby]. Each day they go to the park after school. ans:
↪   (Bobby) is (Aaron)s brother
new:Aaron = "Aaron";
new:Bernardo = "Bernardo";
new:Bobby = "Bobby";
new:Tracy = "Tracy";
(Bernardo) is (Aaron)s brother & (Bobby) is (Bernardo)s brother &
↪   (Tracy) is (Bobby)s mother & (Bobby) is (Tracy)s son & (Bobby)
↪   is (Aaron)s ($relation);
relation-->"#FILE(SCREEN)";
...
```

## K  SYMBOLIC TASKS AT DIFFICULTY LEVEL A (SOURCE CODE)

```
#load(quadratic) // Load the CoL DSL library for Symbolic Tasks
new:x = 1;
6*$x^2 == 3*x - 7;
x-->"#FILE(SCREEN)";
($x - 6)*(x + 3) == x;
x-->"#FILE(SCREEN)";
...
```

## L  SYMBOLIC TASKS AT DIFFICULTY LEVEL B (SOURCE CODE)

```
#load(quadratic) // Load the CoL DSL library for Symbolic Tasks
new:x = 1;
$x*($x + 11) == 16*($x + 22);
x-->"#FILE(SCREEN)";
$x*(36*$x + 50) - 11*(19 - 30*$x) == $x^2;
x-->"#FILE(SCREEN)";
...
```

## M  CROSS-TYPE TASKS

```
#load(quadratic) // Load the CoL DSL library for Symbolic Tasks
#load(family) // Load the CoL DSL library for Relational Tasks
new:x = 1;
$x^2 - 4*$x == 6;
x --> "#FILE(SCREEN)";
...
new:relation = "";
// [Dolores] and her husband [Don] went on a trip to the
↪   Netherlands last year. [Joshua] has been a lovely father of
↪   [Don] and has a wife named [Lynn] who is always there for him.
// Ans: (Dolores) is (Lynn)s daughter-in-law
new:Lynn = "Lynn";
new:Joshua = "Joshua";
new:Don = "Don";
new:Dolores = "Dolores";
(Joshua) is (Lynn)s husband & (Don) is (Joshua)s son & (Dolores)
↪   is (Don)s wife & (Dolores) is (Lynn)s ($relation);
relation-->"#FILE(SCREEN)";
...
```

# N COOL INTERMEDIATE REPRESENTATION

```
"codeTable": [
    {
        "boundtfdomain": "",
        "grounded": false,
        "operand1": {
            "argName": "x",
            "argType": "identifier",
            "changeable": 1,
            "className": "",
            "isClass": 0
        },
        "operand2": {
            "argName": "2",
            "argType": "number",
            "changeable": 0,
            "className": "",
            "isClass": 0
        },
        "operator": {
            "argName": "^",
            "argType": "other"
        },
        "result": {
            "argName": "1418.4",
            "argType": "identifier",
            "changeable": 1,
            "className": "",
            "isClass": 0
        },
        "root": false
    },
    ...
]
```

