# OpenReview forum: "COOL: Chain-Oriented Objective Logic with Neural Networks Feedback Control for Multi-DSL Regulation"
_ICLR.cc/2026/Conference — ICLR 2026 Conference Withdrawn Submission_

### Official Review · Reviewer_JQLx · 2025-10-21

**Soundness:** 1
**Presentation:** 1
**Contribution:** 2
**Rating:** 2
**Confidence:** 3

**Summary:**

The paper studies 'multi-domain-specific languages regulation' using a new neurosymbolic method.

**Strengths:**

- I am not aware of earlier research in this topic
- Ablation studies show clear improvements

**Weaknesses:**

- The problem of Multi-DSL regulation is not introduced, and the writing is very hard to follow: Method description is high level and not reproducible.
- Theoretical claims are unclear and do not come with obvious assumptions. Proofs are often very informal
- The paper does not use baselines
- No code is available

**Questions:**

- Except for the appendix, this paper does not cite any other literature. The problem and approach is therefore not grounded in existing work.
    - For example, how could this paper be tackled using much simpler methods, eg involving LLMs?
    - The citations in the appendix are also suspicious. Eg, [1] is used as a reference for GNNs, but [1] does not use GNNs at all, and, I quote, "Sagar Imambi, Kolla Bhanu Prakash, and GR Kanagachidambaresan. Pytorch. Programming with TensorFlow: solution for edge computing applications, pp. 87–104, 2021" is used as a reference for Pytorch (?). The most similar hit on Google for that reference is a book about Tensorflow.
- The problem ('regulation of multiple modular DSLs') is not introduced and is not clear to me.
- Table 2: What does "Group DSL" mean? Is that a baseline?
- The abstract ends with "(need revision)", and there are many invalid references
- The figures are very hard to parse and do not provide useful insight into the method
- Line 054 claims that 'neural adaptive control (...) is confined to continuous states spaces (...) [and] inapplicable to the discrete symbolic reasoning required'. Neural control can certainly be used for discrete state spaces, in fact this is very common. This claim also comes without support.


Furthermore, I don't like saying this, and I hope I'm wrong, but the writing sounds very LLMy to me. No LLM usage is declared in the paper (which is a requirement by ICLR standards).

[1] Drori, Iddo, et al. "A neural network solves, explains, and generates university math problems by program synthesis and few-shot learning at human level." Proceedings of the National Academy of Sciences 119.32 (2022): e2123433119.

---

> ### Author Response · Authors · 2025-11-29
>
> Apologise for the work not being well delivered due to the limited time (I submitted it just at the deadline in a rush, leaving many details to be improved, such as citations. And your feelings are right, I have used LLMs to compress the content and declared the LLM usage in the submission info, but I didn't notice the code has been updated(I also need to declare LLM usage in the paper). I am not writing to get a higher score, but I really want to know:
>
> 1. If, as long as I use networks (even very lightweight networks for less energy cost, or trained with private knowledge), must I compare them with LLMs? I really don't understand why using neural networks has to be compared to LLMs?  DSL can be used for professional reasoning, especially for those local, small but deep problems, so LLMs cannot replace them (LLM has overshadowed almost all research on reasoning, but it really has nothing to do with my research, and I always get low scores for the doubts about the practical value of the research).
>
> Here are the answers to your questions:
>
> 2. There are no previous works for multi dsl regulation (I have clarified in the paper). People just call the libs together with low robustness. What I have improved is to equip each DSL with an auto-created small network brain, and make DSLs collaborate stably with much less conflict and space to explore. The networks are automatically, continuously, and asynchronously trained; the core of the paper is how to keep the NN system stable.  So the experimental group is how DSLs are differently organised.
>
>
> 3 “Research in neural adaptive control (e.g., in robotics or process control) offers learning-based
> dynamic adaptation but is confined to the continuous state spaces of physical systems, making it inapplicable to the discrete, symbolic reasoning required for coordinating multiple DSLs.”
> It means that neural adaptive control, just like robotic control, which operates in continuous spacetime (based on fixed motion rules) (maybe use "consistent" will be better ), differs in state between consecutive steps only in numerical values, not scale/dimension. MultiDSL controls the switching of rules, using different rules for each step of the inference process (this is the difference between neural adaptive control and neural guided research). Our multi-DSL regulation essentially combines neural adaptive control and neural guided research in symbolic reasoning, making the inference process both flexible and stable. (This definitely needs further clarification.)

---

### Official Review · Reviewer_7Whn · 2025-10-27

**Soundness:** 1
**Presentation:** 1
**Contribution:** 1
**Rating:** 2
**Confidence:** 2

**Summary:**

This paper proposes COOL, a neural-symbolic framework to coordinate reasoning across multiple Domain Specific Languages (DSLs). The framework involves two components:
1. Chain-of-Logic (CoL): a symbolic control mechanism that introduces heuristic vectors and runtime keywords (return, logicjump(n), abort) to manage rule application and cross-DSL transitions.
2. Neural Network Feedback Control (NNFC) – a neural module that employs small neural agents to monitor symbolic derivations, filter errors, and adjust heuristic parameters adaptively.

The authors provide theoretical analyses claiming CoL retains expressiveness of the most powerful DSL, proves complexity reductions, and uses a Lyapunov-based argument to claim stability for NNFC.

Experiments are conducted on synthetic “relational” and “symbolic” benchmarks collected by the paper, showing large performance gains (up to 100% accuracy, 95% faster reasoning, 91% fewer tree operations).

**Strengths:**

- The paper contains many technical details, proofs, and analysis. While I cannot fully understand the work, I acknowledge the effort from the authors.
- The accuracy improvement of the proposed COOL seems to be significant.

**Weaknesses:**

1. **Lack of clarity:** The paper is very difficult to follow. The writing is also hard to understand, and many explanations are very high-level and read like LLM-generated. The explanations on each proposed components are also very high level and lack of implementation details. Here are some questions:
    - How are heuristic vectors constructed?
    - How are the CoL keywords determined?
    - Regarding the neural components, how do you handle non-differentiable operations? Or do you somehow gather ground-truths for each intermediate steps (so you do not back propagate gradients through symbolic/non-differentiable ops)? How large is the training set? How well does it generalize to OOD tasks?
    - The paper uses "agents" a lot. Is it just referring to the NN as in Figure 6? Or does it have anything to do with LLM-based agents?
2. **Poor grounding in existing literature:** The paper is poorly grounded in existing literature. There is no existing works, and no comparison of the proposed approach with existing works. There are a few references in the bibliography, but they are not really cited or referred in the text.
3. **Empirical validity:**
   - Benchmarks are not compared with any existing works.
   - Results show 100% accuracy and 95% speed-ups. I somewhat doubt about the validity of the test data and baseline.

Overall I find the paper very difficult to comprehend, which affects my judgements. I highly recommend the authors to refer to other published papers in similar domains (e.g. [1]) to rework the paper to be more readable.

[1] Latent execution for neural program synthesis beyond domain-specific languages

**Questions:**

1. Could the authors provide more details about the related work and better compare the proposed COOL with existing works?
2. Could the authors provide more implementation details? For instance, how CoL/heuristic vectors are constructed, how keywords are determined, and how are the neural components trained?
3. Could the authors provide evaluation on benchmarks used in existing literature and compare with existing works? For instance, for math problems you can consider e.g. GSM8K, and for logic reasoning please consider various Knowledge Graph Completion tasks (e.g. FB15k, WN18RR etc.)

---

> ### Author Response · Authors · 2025-11-29
>
> Apologise for the work not being well delivered due to the limited time (I submitted it just at the deadline, and I have used llms to compress the content). I am not writing to get a high score, but I really want to know:
>
> 1. If, as long as I use networks (even very lightweight networks), must I compare them with LLMs? If I use the dataset you recommend, will I be asked to show better results compared with other systems that have no relationship with multi DSLs? Then if I failed to provide any soa result, the work would be useless? If it is not allowed to use any self-created dataset?
>
> Here are the answers to your questions:
>
> 2. There are no previous works for multi dsl regulation (I have clarified in the paper). People just call the libs together with low robustness. What I have improved is to equip each DSL with an auto-created small network brain, and make DSLs collaborate stably with much less conflict and space to explore. The networks are automatically, continuously, and asynchronously trained; the core of the paper is how to keep the NN system stable.
>
> 3 I think the experiments are enough to prove the strategy's efficacy, because DSLs are designed for certain kinds of narrow problems, and I don't need datasets that can be used in a wide range. Besides, using self-created datasets in DSL experiments is not an isolated case.

---

### Official Review · Reviewer_sutB · 2025-11-01

**Soundness:** 2
**Presentation:** 1
**Contribution:** 1
**Rating:** 0
**Confidence:** 3

**Summary:**

The work contributes to the management of multi domain specific language (DSL) regulation. It introduces 1) Chain of Logic (CoL) and a neural network feedback control (NNFC). Chain of Logic uses heuristic vectors and runtime keywords as syntax to specify what DSL is suitable for that part of a text (or program?). The neural network feedback control introduces "neural agents", which may adaptively redefine the scope of what a DSL is responsible for.

**Strengths:**

The work claims positive results. (Although the experimental section does not explain exactly what task exactly is being tested on).

**Weaknesses:**

The presentation is unsuitable for a broader audience. The exact task that is being solved is not explained, neither what task is performed in the experiments. As a consequence, the contribution itself is also not clear. The main paper is not self-contained.

The main paper contains no references. The few references that are included originate from the appendix. For example, there are no citations to prior work (i.e., line 051 states intersections with several research areas are fundamentally distinct. No citations appear).

The work is not well motivated, by not having a clear description of the problem and a description of its importance with references to prior work.

**Questions:**

Regarding more specific suggestions:

* Figure 1 is not self contained, and from the text it is unclear what it is showing exactly.

* Line 237 has a broken Appendix reference

---

### Official Review · Reviewer_RwiZ · 2025-11-01

**Soundness:** 1
**Presentation:** 1
**Contribution:** 1
**Rating:** 0
**Confidence:** 4

**Summary:**

This paper proposes Chain-Oriented Objective Logic (COOL) as a neuro-symbolic framework for Multi-DSL regulation. To achieve this, the authors introduce Chain-of-Logic (CoL) and Neural Network Feedback Control (NNFC) to create a hierarchical reasoning process with self-correcting mechanisms.

**Strengths:**

Due to the manuscript's lack of clarity and the absence of a discussion on related work, I am unable to confidently identify the strengths of this work.

**Weaknesses:**

- Although the authors claim "Our approach to multi-DSL regulation, COOL (Chain-Oriented Objective Logic), intersects with several research areas" (line 50), the paper does not contain a related work section.
- All 12 references to related work the paper contains are contained in the appendix, i.e., the main text does not contain a single reference as far as I can tell.
- The paper contains erroneous LaTeX commands (e.g., line 237 "Apendix ??") and unfinished text fragments (e.g., "(need reversion)" comment in line 28).
- Neither the text nor the figures enabled me to understand the proposed approach, and rather left me with more questions than answers.

**Questions:**

I have no questions to the authors.

---

### Note · Authors · 2025-12-02

I have read and agree with the venue's withdrawal policy on behalf of myself and my co-authors.